# The Oxygen Cascade from Atmosphere to Mitochondria as a Tool to Understand the (Mal)adaptation to Hypoxia

**DOI:** 10.3390/ijms24043670

**Published:** 2023-02-12

**Authors:** Michele Samaja, Sara Ottolenghi

**Affiliations:** 1MAGI GROUP, San Felice del Benaco, 25010 Brescia, Italy; 2School of Medicine and Surgery, University of Milano Bicocca, 20126 Milan, Italy

**Keywords:** hypoxia, altitude, adaptation, maladaptation, genome-wide association studies

## Abstract

Hypoxia is a life-threatening challenge for about 1% of the world population, as well as a contributor to high morbidity and mortality scores in patients affected by various cardiopulmonary, hematological, and circulatory diseases. However, the adaptation to hypoxia represents a failure for a relevant portion of the cases as the pathways of potential adaptation often conflict with well-being and generate diseases that in certain areas of the world still afflict up to one-third of the populations living at altitude. To help understand the mechanisms of adaptation and maladaptation, this review examines the various steps of the oxygen cascade from the atmosphere to the mitochondria distinguishing the patterns related to physiological (i.e., due to altitude) and pathological (i.e., due to a pre-existing disease) hypoxia. The aim is to assess the ability of humans to adapt to hypoxia in a multidisciplinary approach that correlates the function of genes, molecules, and cells with the physiologic and pathological outcomes. We conclude that, in most cases, it is not hypoxia by itself that generates diseases, but rather the attempts to adapt to the hypoxia condition. This underlies the paradigm shift that when adaptation to hypoxia becomes excessive, it translates into maladaptation.

## 1. Introduction

About 1% of the world human population live permanently at >2500 m altitudes, mainly in non-Western countries [1]. The major health problem these people constantly face stem from breathing an atmosphere impoverished in oxygen (O_2_). On average, up to 10% of the population living at altitude develop diseases linked to insufficient O_2_ or hypoxia [2]. Hypoxia may occur even at sea level, contributing to morbidity and mortality in numberless patients with acute respiratory diseases (ARDS), chronic pulmonary diseases (COPD), emphysema, lung tumors, anemia, heart failure, and hundreds other disorders that stem from unmatched O_2_ supply with respect to needs. Such diseases take an intolerable toll on humanity in terms of sufferance, life loss, and economic distress: the burden imposed in one single year (2010) in the US only by COPD alone was USD49.9 billion [3]. Among all the factors that afflict humanity, hypoxia has the unique feature of being both physiological (i.e., due to altitude), and pathological (i.e., due to a pre-existing disease). When physiological, hypoxia may raise some form of adaptation, and the ability to adapt to environmental stress factors represents a key feature in the Darwinian theory of evolution of species. However, it is not yet clear whether humans can ever adapt to hypoxia, given the high incidence of the clinical manifestations associated with both physiological and pathological hypoxia.

The aim of this review is to focus on the capacity to adapt to the challenge represented by insufficient O_2_. To this purpose, we will examine the mechanisms underlying the responses to hypoxia at several levels spanning from genes to the respiratory and circulatory features, focusing on those that are most relevant in terms of adaptation. To accomplish this goal, we will reexamine the O_2_ cascade as a tool to understand the genesis and consequences of physiological and pathological hypoxia. We will focus on the paths that impact hypoxia adaptation neglecting the contributions that describe the simple responses to hypoxia. When possible, we will privilege contributions referring to humans rather than animals or in vitro. To indicate the degree of hypoxia, we will use the double notion of altitude in meters above sea level and the corresponding effective %O_2_, where 21.0 %O_2_ is equivalent to sea level.

## 2. Oxygen and Hypoxia

Three 18th century scientists participated in the discovery of O_2_. Carl Wilhelm Scheele (1742–1786), a Swedish–German pharmaceutical chemist, was the first to discover O_2_, but his low academic profile and lack of international competitiveness caused him the loss of priority for this discovery [4]. Joseph Priestley (1733–1804), a polyhedric English chemist, natural philosopher, separatist theologian, grammarian, multi-subject educator, and liberal political theorist, was able to intertwine his scientific work with the ongoing debate on *phlogiston* within a conflict with the Church of England that led him to migrate to the US. His publication, although delayed with respect to that of Scheele, gained him the priority as discoverer of O_2_ [5]. Finally, Antoine Lavoisier (1743–1794), a French nobleman and chemist, coined, together with Marie-Anne Lavoisier, the term “oxygen”, based on the misleading concept of a substance that generates acidity (*-ὀξύς* (*oxys, acid*) *+ -γενής* (*-genēs*, *generator*)), during the years of the French Revolution that led Antoine to execution under the guillotine [6]. The contributions by two other 17th century scientists, the English John Mayhow (1661–1679) and the Polish Michael Sendivogius (1566–1636) should also be mentioned, although their extraordinary work probably was not robust enough to assign them the role of discoverers of O_2_ [7].

### 2.1. Oxygen at Altitude

Dalton’s law enables the expression of the partial pressure of O_2_ (PO_2_) as a function of %O_2_, the barometric pressure (BP), and the aqueous vapor pressure (P_H2O_):(1)$${\mathrm{PO}}_{2}=\frac{\left(\mathrm{BP}-P_{H2O}\right)}{100}\times {\%O}_{2}$$

BP and altitude are the main determinant of atmospheric PO_2_ (Figure 1), as P_H2O_ depends uniquely on temperature (47 mmHg at 37 °C) and %O_2_ is remarkably constant (20.9460%) at any latitude on Earth, except Antarctica, where physical factors slightly decrease the %O_2_. At Dome C Concordia, which is located at a geographical altitude of 3233 m/14.3 %O_2_, the slightly lowered %O_2_ in air (20.82–20.90) [8] decreases the effective %O_2_ to 3800 m/13.2% [9].

Decreasing the %O_2_ in breathed air has dramatic effects on exercise capacity. Figure 2 shows the resting and maximal O_2_ uptake (VO_2_) with altitude. The resting VO_2_, not affected by altitude, remains ~3.5 mL O_2_/kg/min throughout, or 1 metabolic equivalent of training (MET). The maximal VO_2_, by contrast, decreases progressively from ~18 MET in exceptionally well-trained subjects at sea level to 3–4 MET near the summit of Mt. Everest (8848 m/6.6 %O_2_), which corresponds to activities such as walking, machine tooling, standing tasks, or golfing. Thus, decreased O_2_ availability at altitude would enable subjects to perform very light tasks only—let alone climbing and fighting the extremely low temperatures at those altitudes. As the incidence of resting VO_2_ on maximal VO_2_ increases from 5% at sea level to ~30% at 8848 m/6.6 %O_2_, at extreme altitudes a significant portion of inhaled O_2_ is diverted just to sustain the basal needs of the organism. The first BP measurement taken on the Mt. Everest summit at 8848 m was 253 mmHg [10], perhaps sufficient to sustain basic metabolism. However, measurements taken with stratospheric balloons showed relevant BP variability depending on latitude and seasons [11].

### 2.2. Hypoxia

Hypoxia is defined as an imbalance between O_2_ supply and O_2_ demand. This definition applies at all organization levels from whole organisms to cells in both physiological and pathological contexts. Although the basic concepts related to altitude hypoxia are now found in several human diseases, until recently hypoxia was recognized almost exclusively in altitude contexts, in inhabited balloons flights, as well as in occasional or permanent dwellers in the mountains (Table 1).

Although the hypoxia severity may range from mild to severe depending on arterial PO_2_ and exposure duration [15], the threshold PO_2_ that triggers the hypoxia condition is rather undefined as it depends on pathophysiological factors such as the cell metabolic rate, the VO_2_, the cardiac output, and the O_2_ extraction, just to name a few, that cause extreme variability in the threshold PO_2_. Some patients with COPD, heart failure, and anemia are indeed hypoxic even at 0 m/21.0 %O_2_, while healthy untrained subjects may dwell for short times at >3300 m/14 %O_2_ without developing appreciable signs of sickness. By contrast, extremely well-trained subjects may survive at 7500 m/8.0 %O_2_ for hours/days. As for cells, current good practice [16] suggests exposing cultured cells to >20,000 m/1.0 %O_2_ to simulate hypoxia.

Even the various organs in the same organism vary remarkably in terms of VO_2_, capillary blood flow, and O_2_ extraction. This translates to different sensitivities to the same degree of hypoxia depending on the specific organ’s physiological status. In rats exposed to the same degree of hypoxia, various organs respond differently in terms of triggering the O_2_-sensing machinery and of activation of apoptosis [17]. It is thus expected that at every degree of hypoxia, the organs with a high metabolic rate, such as the brain, heart, and exercising muscles, become hypoxic before those with a low metabolic rate, where the O_2_ consumption is better matched with perfusion.

### 2.3. Clinical Manifestations of Hypoxia

How to establish if an organism becomes adapted to a challenge? In principle, wherever the challenged population is free from the diseases inferred by that challenge, can survive, and can reproduce, then it may be considered adapted. When hypoxia constitutes the challenge, the diseases may be acute or chronic, giving rise to the complex pathology addressed by acute (AMS) and chronic mountain sickness (CMS), respectively. Up to 10% of the world population living at altitude develop CMS [2,18], although with marked geographical differences.

The onset of CMS, first described by Carlos Monge Medrano [19], is considered a valuable marker of the adaptation to chronic hypoxia. An elusive disease caused by sustained insufficiency of O_2_, its major pathogenic etiology is the excessive erythropoietic response, or polycythemia, which originates outcomes collectively known as CMS [20]. Thus, tracking hematological changes enables monitoring of hypoxia adaptation [21,22]. Alternate valid markers of maladaptation include the development of pulmonary hypertension (PH), a frequent condition at altitude, which may be accompanied by right ventricle hypertrophy and failure. Neuropsychological dysfunctions are also of concern in CMS as they may affect school performance in children [23]. To partially compensate such effects, the “oxygen conditioning” paradigm was proposed to enrich %O_2_ with the use of synthetic zeolite, which adsorbs air nitrogen, based on the observation that a 1 %O_2_ increase corresponds to a decrease in altitude of 300 m [24].

## 3. The Oxygen Cascade

Despite important limitations [25], the O_2_ cascade remains a good representation of the path of O_2_ from atmosphere to mitochondria, the final O_2_ users (Figure 3). The blue line represents the progressive stepwise decreases in PO_2_ along the path in normoxic healthy conditions. The establishment of O_2_ gradients at the interface between adjacent compartments ensures the correct flow of O_2_ along the cascade. Each of these gradients must be wide enough to grant a sufficient O_2_ supply to the mitochondria [26]. The red line represents the situation established when atmospheric PO_2_ is reduced by one-third, equivalent to 3400 m/13.9 %O_2_. Such a drop translates into a reduced width of each gradient step, which corresponds to a decreased supply of O_2_ to the mitochondria. Thus, all the interfaces must cooperate to ensure the correct O_2_ supply. The same reduction in O_2_ supply to mitochondria occurs when the environmental PO_2_ is normal, but some sites of resistance to the O_2_ flux emerge in the correspondence of one of the interfaces. This situation corresponds to wider PO_2_ gradients between adjacent compartments. Such sites of resistance are usually associated with diseases that transform a normoxic situation into a hypoxic one downstream of the block in the O_2_ cascade. This determines the establishment of pathological hypoxia, which differs from the physiological one by being often accompanied by factors such as inflammation and redox imbalance, just to name a few.

The ability to adapt to environmental changes is a key feature in the evolution of species. While the mechanisms underlying the cardiac, nervous, and pulmonary tissues’ initial adaptation, or acclimatization, to hypoxic conditions have recently been worked out [27], the rules for recognizing the onset of long-term adaptation should include the appearance of complex characters that are too well-fitted to the environment for the fit to have arisen by chance and that help their bearers to survive and reproduce. The concept of adaptation physiology greatly depends on semantic problems related to the meaning of this term [28]. Although adaptation is believed to involve the establishment of compensatory responses to environmental challenges, not all the responses to a challenge may be adaptive. Thus, there may be three levels of response to challenges depending on the gain: inadequate (gain < 100%), ideal (gain = 100%), and excessive (gain > 100%) (Figure 4). This review will analyze the responses to the hypoxic challenges in the various resistance sites represented in Figure 3 in the search of an answer to the question of whether humans can ever adapt to hypoxia.

### 3.1. Cell-to-Mitochondria Interface

#### 3.1.1. Hypoxia-Inducible Factors

Understanding how cells sense and adapt to O_2_ availability was worth the 2019 Nobel Prize in Physiology and Medicine to William Kaelin, Peter Ratcliffe, and Gregg Semenza, as well as robust worldwide reconnaissance. This ubiquitarian mechanism constitutes the O_2_-sensing function that orchestrates most, although not all, the cell responses to O_2_ shortage. The transcription factor family of hypoxia-inducible factors (HIF), a fundamental part of the O_2_-sensing system, exploits its function by inducing or repressing the expression of hundreds of genes encoding for proteins that target a relevant number of metabolic (e.g., glycolysis), morphological (e.g., angiogenesis), cell proliferation and survival (e.g., apoptosis, cell cycle control), and molecular (e.g., production of erythropoietin) processes.

HIF activity is only in small part regulated by gene expression, with the bulk of the control entrusted to prolyl hydroxylases (PHD) and the factor inhibiting HIF (FIH), which, in the presence of O_2_, hydroxylate HIF at either the two prolyl or the asparaginyl residues, respectively [29]. When hydroxylated, HIF undergoes ubiquitination and cannot be active in its task of controlling the gene expression. While the β subunit that composes HIF is constitutive and does not respond to O_2_, at least three isoforms of the O_2_-sensitive α subunit have been identified. The best known of them, HIF-1α and HIF-2α, share 48% sequence homology but are differentially expressed in various tissues. For example, HIF-2α, also known as the endothelial PAS domain protein 1 (EPAS-1), is principally expressed in endothelial cells [30], brain, heart, lung, kidney, liver, pancreas, and intestine tissues [31]. HIF-2α regulates fewer genes than HIF-1α yet plays an important role in erythropoiesis [32,33]. Whereas HIF-1α appears to be more expressed in response to short periods of intense hypoxia, HIF-2α seems more active during prolonged mild hypoxia [34].

The regulation of HIF activity as a function of O_2_ availability relies on the sensitivity of PHD and FIH to PO_2_ changes, which is expressed by the value of the Michaelis–Menten constant (K_M_), or the PO_2_ at which the reaction rate is half-maximal. Given the average intracellular PO_2_ in the 7–20 mmHg/1.0–2.5 %O_2_ range, FIH (K_M_ = 50–80 mmHg/6.5–10.5 %O_2_) appears dynamically more effective than PHD (K_M_ = 120–210 mmHg/15.7–27.6 %O_2_) as a physiological O_2_ sensor [29,35,36]. However, K_M_ values are to be taken with caution because they are markedly affected by the length of the substrate peptides necessary to perform the measurement. In addition, the slope of the curve representing HIF activity vs. PO_2_ needs to be assessed to fully appreciate the role of HIF in hypoxia adaptation, but no data are yet available on this key information.

Understanding the rules of the O_2_-sensing mechanisms of HIF and appreciating HIF as a key factor for hypoxia adaptation requires not only the availability of kinetic data addressing the O_2_ affinity of PHD and FIH, but also HIF stability in prolonged and chronic hypoxia. In vitro experiments, mostly performed in cultured tumor cells because the problem of prolonged hypoxia is particularly compelling in cancer growth contexts, may help address this issue in part, but data are quite contradictory. In neuroblastoma cells, both HIF-1α and HIF-2α proteins become destabilized after >72 h hypoxia [37]. It was proposed that during cancer development, acute hypoxia first stabilizes HIF-1α, which decreases at about 24 h and is replaced by HIF-2α, which in turn slowly decreases and is replaced by other hypoxia-associated factors [38]. One study followed human microvascular endothelial cells during 14-day hypoxia, highlighting the messenger RNA (mRNA) expression response associated with HIF-2α as more prolonged than that associated with HIF-1α, but HIF protein data, which are more suitable to address the role of HIF in hypoxia adaptation, are lacking [39]. Acute and chronic hypoxia can lead to completely different hypoxia-related responses. While acute hypoxia affected the expression of 196 genes, chronic hypoxia resulted in changes in 4149 transcripts, with only 144 genes common to both [40]. Such marked changes in the transcriptome, potentially able to address the role of HIF in hypoxia adaptation, may stem from the activation of different downstream transcriptional effects originating from the various HIF isoforms [41]. In facts, the mRNA expression of the HIF isoform HIF-3α during 48 h hypoxia revealed that while the HIF-1α mRNA peaked at 4 h, HIF-3α mRNA expression increased on HIF-1α decrease [42].

Spontaneous PO_2_ fluctuations in tumors in vivo further complicate assessing the role of HIF in prolonged hypoxia, and thus its involvement in hypoxia adaptation, also due to the great heterogeneity in experimental protocols for hypoxia levels and exposure times, which may range from 30 min to several weeks [43]. In healthy rats exposed to 5900 m/10.0 %O_2_ chronic hypoxia without reoxygenation for 14 days, HIF-1α protein increased in the brain [44,45], muscle [46], and heart [47], but heart mRNA was unaffected [48].

In summary, as HIFs may regulate the expression of several hundred genes, perhaps up to 2.6% of the human genome [49], it remains difficult at present to assess if all such activities translate to useful hypoxia adaptation patterns, leaving the question whether HIFs participate to the adaptation processes relatively unanswered.

#### 3.1.2. Antioxidant Defense

Besides fueling oxidative phosphorylation, O_2_ may also form highly reactive species (ROS) either through non-enzymatic (e.g., mitochondrial uncoupling) or enzymatic [50] processes. When elevated, ROS are toxic due to their high reactivity, but at physiological levels they modulate a wide range of pathways that control virtually all cell functions. ROS are directly linked to the genesis of oxidative stress, which is the outcome of the imbalance between aggressive species formation and antioxidant defenses.

Mitochondrial uncoupling is the main, but not exclusive, way through which hypoxia causes ROS formation [51]. The crosstalk between dysfunctional mitochondria and NADPH oxidase in the plasma membrane [52] engages this enzyme as an additional ROS source [53]. NADPH oxidase 4 responds directly to the O_2_ lack [54], thus constituting a major ROS source in hypoxia [55,56], and nearly doubles in the brains of mice exposed to 5900 m/10.0 %O_2_ for 4 weeks [57]. NADPH oxidase 4 is antagonized by the nuclear factor erythroid 2-related factor 2 (Nrf2), which in rat brains responds to 12 h hypoxia [58] and upregulates the antioxidant defenses [59]. Another protective factor, protein kinase B or Akt, is a serine/threonine-specific protein kinase that plays a key role in neuroprotection by inhibiting apoptosis [60,61] and revealed protective features even in chronic situations, possibly overriding the protection elicited by Nrf2 [57]. Mild 5900 m/10.0 %O_2_ hypoxia increases both ROS production and defense mechanisms, but since the first is stronger, at least for 4-week hypoxia, the redox imbalance results worsened with the onset of clear signs of oxidative stress [57].

The antioxidant glutathione, a tripeptide thiol that participates to the cell defense against ROS, may also play a relevant role. The ratio of oxidized/reduced glutathione (GSSG/GSH) reflects the cell redox status [62], and the high vulnerability of neurons to ROS may be linked to the relatively low activity of glutathione peroxidase in the brain [63]. As breathing at 8500 m/7.0 %O_2_ for 6 h reduces GSH and glutathione peroxidase activity in brain extracts, hypoxia clearly impairs the antioxidant defense constituted by glutathione [64].

Exposure to altitude is widely known to enhance ROS generation that overwhelms the cell defenses and leads to lipid, protein, and DNA damage, thereby causing or exacerbating the outcome of several diseases including CMS [65]. The adaption to the oxidative challenge requires relatively long periods of time, with physical exercise usually exacerbating the severity of this challenge [66]. In fact, in humans exposed to 3800 m/13.2 %O_2_ the indirect markers of oxidative stress peak by the 20^th^ day of hypoxia, followed by a sustained increase for 10 months [67].

Neurons and the brain are highly vulnerable to ROS [68,69] because of high VO_2_ and low antioxidant defense [70]. Consequently, one of the most remarkable side effects of hypoxia-induced oxidative stress is the frequently observed phenomena related to memory and cognitive impairment. This outcome was observed not only in neurological studies [71], but even in animal models [72]. Thus, the cognitive impairment may be directly linked to the onset of oxidative stress caused by insufficient antioxidant buildup as the increased ROS levels in rats exposed to 6100 m/9.8 %O_2_ for 7–14 days correlated with nNOS expression, neurodegeneration, and DNA fragmentation in the hippocampus, cortex, and striatum [73]. Likewise, 2-week 5900 m/10.0 %O_2_ hypoxia was reflected in marked neuronal apoptosis, a hallmark of overt brain damage [44,45].

In summary, there are two aspects to address the ox–redox challenge: the increased hypoxia-induced oxidative stress and the buildup of endogenous antioxidant defenses, which might not be fully exploited, especially for longer hypoxia durations. In terms of hypoxia adaptation, it might appear that the potentially adaptive response represented by the antioxidant defenses is insufficient to meet its target.

#### 3.1.3. Bioenergetics

The great majority, perhaps 97%, of breathed O_2_ is employed in the final steps of long chains of reactions that oxidize substrates, mainly glucose, fatty acids, and in certain organs, ketone bodies (Figure 5). The O_2_ vehiculated in the cell, either through simple diffusion or, in contractile cells, via binding with myoglobin (Mb), is finally used up by cytochrome oxidases to yield biological energy in the form of adenosine triphosphate (ATP). We neglect here the contribution of alactic anaerobic pathways, i.e., the biological energy derived from the phosphocreatine shuttle and the adenylate kinase reaction, which provide ATP only for limited periods of time. Clearly, any change in mitochondria morphology and density is unavoidably reflected in changes in the bioenergetic paths.

The effect of hypoxia on the mitochondrial volume and density in skeletal muscle is controversial, mainly due to the presence of two superimposing effects: exercise training, which is expected to increase mitochondrial volume, and hypoxia, which instead antagonizes the effects of training with respect to mitochondria [74]. Clearly, variables linked to hypoxia duration and intensity and to the degree of exercise come into play [75]. However, animal studies have shown that the activity of muscle cytochrome oxidase falls markedly in both chronic and acute hypoxia, which depresses energy production by the mitochondria [76]. In humans, in situations not disturbed by the acute onset of exercise programs, lower muscle mitochondrial contents have been reported in Sherpas [77] and lowland-dwelling Tibetans [78] compared with lowlander populations. Furthermore, the capacity for fatty acid oxidation is reduced in altitude Sherpas, due to downregulated muscle expression of the gene that encodes peroxisome proliferator-activated receptor alpha, a transcriptional regulator of fatty acid metabolism [79]. Decreased mitochondrial volume with a loss in oxidative capacity is indeed a constant finding in physiological hypoxia studies related to sea-level individuals brought to altitude [80].

With impairment of mitochondrial function, glycolysis should surge as pivotal to supplying anaerobic ATP to the cell, the so-called glycolytic switch or Warburg effect. Originally identified in hypoxic cancer cells, the Warburg effect describes the increasing reliance of the cell metabolism on glycolysis to spare the mitochondrial function [81]. The discovery that the Warburg effect is activated by HIF-1α [82] provides the structural basis to identify the glycolytic switch as a response to hypoxia. Although less efficient in building biological energy, glycolysis may nevertheless supply ATP, and human red blood cells (RBCs) rely entirely on glycolysis to produce energy. As well, strongly exercising muscles occasionally rely on anerobic mechanisms, although with lower energy conversion efficiency.

Glycolysis releases lactate and H^+^. Although high H^+^ produces reversible effects in isolated perfused hearts, high lactate causes irreversible damage the O_2_ utilization paths by at least two mechanisms: (*a*) impairment of phosphorylation coupling efficiency due to the dissipation of the H^+^ gradient across the inner mitochondrial membrane via upregulation of lactate–H^+^ cotransport, and (*b*) lactate-induced increased amplitude of the Ca^++^ transients, which leads to an increased intracellular Ca^++^ load with higher costs required for pumping out Ca^++^ [83]. The limiting factors for lactate exportation out of the cell will be discussed below. Exported lactate is then re-converted to glucose by liver gluconeogenesis at the cost of 6 ATP/mole, while converting glucose to lactate in the glycolysis yields only 2 ATP/mole. Actually, however, the liver ATP necessary for this process, or the Cori cycle, which compensates for the side-products of anaerobic mechanisms in peripheral hypoxic cells, is produced by aerobic mechanisms, which addresses O_2_ as a non-replaceable substrate for life. The intracellular lactate buildup consequent to the Warburg effect acidifies the cell milieu to a much larger extent than CO_2_, the product of oxidative phosphorylation. Increased cell acidity conveys side effects that are usually considered noxious. The Warburg effect is commonly observed in carcinogenesis, as wells as in hypoxic lungs in all their components, i.e., smooth muscle cells, endothelial cells, and fibroblasts [84]. Studies of human PH [85] as well as a meta-analysis of PH experimental models [86] confirmed that the changes in substrate utilization in the right ventricle that correlate with disease severity are compatible with the insurgence of the Warburg effect.

Recently, the favorable effect of erythropoietin (EPO) on mitochondrial functionality was examined in both in vivo and in vitro models of Parkinson’s disease, in which augmented oxidative stress leads to mitochondrial disfunction, neuronal apoptosis, and cell toxicity [87,88]. In these models, the administration of EPO could not rescue the mitochondrial functionality, yet it accelerated the glycolytic rate, contributing to the partial restoration of the ATP levels, i.e., in a way similar to that exerted by hypoxia adaptation [89]. It is, however, a matter of fact that the EPO levels usually peak a few days after the onset of hypoxia but gradually return to baseline in the following weeks, rendering it difficult to identify a role of EPO in protecting the mitochondria from prolonged hypoxia and hypoxia adaptation.

In summary, the bioenergetic changes associated with hypoxia, the logical consequences of the aerobic-to-anaerobic switch, may favor hypoxia adaptation, but the unavailability of methods to directly measure the O_2_ delivery in vivo at the mitochondrial level without the use of indirect mathematical calculations prevents an appreciation of how the bioenergetic changes may impact hypoxia adaptation. In addition, the role of pleiotropic factors that may modulate aerobic and anaerobic energy production, such as EPO, although already hypothesized [90], is still to be worked out.

#### 3.1.4. Lactate Shuttle

An outcome of the Warburg effect, the aerobic-to-anaerobic switch may become costly for cells not only for the impaired energy production, but also because of lactate overproduction. Lactate toxicity may be contrasted, at least in part, by favoring the export of lactate out of the cell. The lactate shuttle hypothesis covers the intracellular and cell–cell movements of lactate produced by glycolysis [91]. While often perceived as a toxic product of anoxic cell metabolism and a determinant of muscle fatigue, lactate mediates several processes spanning from wound repair and regeneration to the mechanisms related to astrocyte–neuron synergy, the lactate–alanine conversion into glucose in the liver via the Cori cycle, and the peroxisome and spermatogenic functions [92].

A key factor in the lactate shuttle is the monocarboxylate transporter (MCT), a ubiquitously expressed family of at least 14 membrane transporters that carry lactate, pyruvate, and ketones across the cell membrane [93]. The most-studied MCT isoforms possess different K_M_ values for lactate [94]: while MCT2 has a high affinity for lactate (K_M_ = 0.5–0.75 mM), MCT1 has a lower affinity (K_M_ = 3.5–10 mM) [95]. MCT4, initially believed to be a low-affinity transporter, was revealed instead to be a high-affinity lactate–H^+^ symport capable of exporting lactate against significant gradients [96].

The role of MCTs in hypoxia stems principally from the observation that high expression of MCTs correlates with tumor aggressiveness and malignancy [97]. In triple-negative breast cancer, increased expression of MCT4 correlates with the clinical outcome [98], while upregulation of MCT1 supports the glycolytic phenotype of glioblastomas [99]. Both MCT1 and MCT4 are critical for the growth of glycolytic tumors [100] and even MCT2 was found to respond to microenvironmental hypoxia and acidosis to maintain a high glycolytic flux in glioblastomas [101]. Such observations addressed the rationale of targeting MCT-mediated lactate transport for the treatment of some types of cancer [102]. Indeed, augmented intracellular acidosis due to MCT4 inhibition with depressed lactate export decreased cancer cell survival [103]. Likewise, targeting MCT1 in endothelial cells inhibited tumor angiogenesis secondary to lactate-induced HIF-1 activation [104]. By binding and stabilizing a protein that triggers signals for cell growth and angiogenesis, lactate may promote a hypoxic response independently of HIF [105]. However, MCTs are also clinically relevant in diseases involving immunosuppression, inflammation, insulin resistance, and brain function. Almost all MCTs are expressed in red gastrocnemius muscle at the mRNA level and their expression is regulated differently under hypoxia preconditioning and exercise conditions [106]. Prolonged hypoxia induces MCT4 expression in mesenchymal stem cells, which results in a secretome that is deleterious to cardiovascular repair [107].

It may be intuitive that the expression of MCTs and lactate dehydrogenase would increase in physiological hypoxia to facilitate the disposal of glycolytic lactate, thereby preventing acidification of the cell milieu. In rats exposed to 4500 m/12.1 %O_2_ for 8 weeks, the response was, however, extremely tissue-specific: MCT4 increased by 34% in hearts but decreased by 47% in plantaris muscle, while MCT1 decreased in plantaris muscle, indicating a minor role for MCTs in hypoxic contractile tissues [108]. By contrast, 48 h hypoxia increased MCT1 mRNA and protein (×8.5 and ×2.7, respectively), as well as MCT4 mRNA (×14.3), but not the protein in human adipocytes in culture [109]. Thoroughbred horses trained for 2 weeks under mild hypoxia (1390 m/18.0 %O_2_) displayed increased MCT4 protein levels and phosphofructokinase activity, vs. constant MCT1 [110]. Horses subjected to more severe hypoxia (3000 m/14.7 %O_2_) did not show changes in MCT1 protein, but MCT4 protein increased by 13%, which highlights that hypoxia may improve exercise performance and glycolytic capacity of skeletal muscle at least in horses [111]. Finally, in rat soleus and EDL muscles, high-intensity interval training, a model of simulated hypoxia, elevated the mRNA levels of MCT4, PGC-1α, and HIF-1α, but no protein data are available [112]. Such data may thus indicate that MCT1 is insensitive to O_2_ changes, while MCT4 is dependent on O_2_ levels.

In summary, in the absence of targeted studies addressing the role of MCT to favor lactate export in somatic cells, there is little support for a relevant role for MCT in hypoxia adaptation.

### 3.2. Capillary-to-Cell Interface

#### 3.2.1. Nitric Oxide

Despite its short half-life in the circulation of a few milliseconds [113,114], NO, a critical mediator of the circulatory responses to hypoxia, explicates its action through vasodilation, which rescues the O_2_ supply/demand ratio that is disrupted by hypoxia, especially in acute scenarios [115]. NO acts as a master regulator of the vascular tone and blood pressure [116]. Other important NO functions are also mediated by nitrosylation of proteins such as HIF-1α, which is then stabilized not only by hypoxia but also by NO-dependent S-nitrosylation [117]. A small free radical, NO is produced in the NO synthase (NOS) reaction, with O_2_ being one of the substrates:2 L-Arginine + 3 NADPH + 3 H^+^ + 4 O_2_ → 2 citrulline + 2 NO + 4 H_2_O + 3 NADP^+^(2)

This reaction may be catalyzed by at least three distinct NOS isoforms, each coded in different chromosomes: (1) the Ca^++^-sensitive, constitutive neuronal NOS (nNOS), located mainly in the nervous tissue and in smooth and skeletal muscles, produces the bulk of the NO that acts as a neurotransmitter; (2) the Ca^++^-insensitive, inducible NOS (iNOS), located mainly in the immune and cardiovascular systems, in vessel walls and macrophages, in response to inflammatory stimuli and hypoxia; and (3) the Ca^++^-sensitive, constitutive endothelial NOS (eNOS), expressed in the endothelium of virtually all organs, is subjected to genetic polymorphism [118,119] and is responsible for smooth muscle relaxation.

The regulation of NOS activity by hypoxia has the ability to fine-tune the O_2_ delivery to cells [120]. The K_M_ value of NOS for O_2_ is the key to appreciate the O_2_ sensitivity of the NO-producing machinery. Briefly, the NO synthesis rate in the brain and endothelium is tightly linked to the onset of hypoxia because the K_M_ for O_2_ of the overall NOS activity (3.6–14.3 mmHg/0.5–1.9 %O_2_ [121]) is near the expected tissue PO_2_, which renders the PO_2_ variations dynamically linked to NO production. The nNOS isoform (K_M_ = 158 mmHg/20.8 %O_2_ [122]) is expected to respond to PO_2_ changes, but with less capacity as the microenvironmental PO_2_ is much less than the K_M_. By contrast, eNOS displays a high affinity for O_2_ (K_M_ = 16.6 mmHg/2.2 %O_2_ and 5.5 mmHg/0.7 %O_2_ for brain and endothelium, respectively [121]), suggesting that eNOS operates in an environment that already saturates the enzyme, unless in extreme hypoxia conditions. Thus, in non-extreme situations, the changes in PO_2_ sensed by eNOS are not expected to be tightly linked to changes in NO production.

After release into the blood stream, NO establishes equilibria with at least three components (Figure 6): (1) an inorganic reservoir constituted by plasma NO_2_^−^ + NO_3_^−^, (2) RBC hemoglobin (Hb) to form nitrosylated or nitrosothiolated Hb [123], and (3) soluble guanylate cyclase in smooth muscle cells, which generates cyclic guanosine monophosphate (cGMP), which induces muscle relaxation through the activation of protein kinase G and reduction in intracellular Ca^++^ [124].

High circulating NO levels are toxic because NO is itself a free radical and its reaction with O_2_ generates peroxynitrite, a most dangerous reactive nitrogen species. This results in nitrosative stress, which plays an important role in the pathology of various diseases such as heart failure [125]. On the other hand, when present at physiological or slightly supraphysiological levels, NO protects the hypoxic body in several ways.

Besides its main function as a vasorelaxant to enhance tissue perfusion and oxygenation, NO restores peripheral blood mononuclear cell adhesion, which is considerably reduced in hypoxia. This function is explicated via an interaction with the mechanisms that affect remodeling and the extracellular matrix. First, hypoxia-induced overexpression of eNOS tends to rescue basal NO levels, which recovers cell–matrix adhesion via cGMP-dependent protein kinase [126]. This favors the cell’s ability to adhere to the extracellular matrix, an essential step to maintain normal homeostasis, which is hampered by hypoxia, which reduces NO bioavailability (see below). In addition, while hypoxia-induced over-perfusion of the pulmonary bed destabilizes the endothelial cells by increasing their leakiness via depolymerization of F-actin stress fibers (which impairs the permeability of the pulmonary vascular endothelial bed, potentially leading to high-altitude pulmonary edema), activating the NO/cGMP pathway with NO donors or sildenafil can revert this process [127].

NO is tightly linked to hypoxia. In acute contexts (hours–days), hypoxia decreases pulmonary NO and causes vasoconstriction, which may ultimately lead to hypoxic pulmonary vasoconstriction [128,129,130]. In patients with high-altitude pulmonary edema, these symptoms are alleviated by inhaled NO with improvement of the arterial O_2_ saturation [131]. By contrast, dietary nitrate supplementation in healthy adult volunteers residing at 4559 m/12.0 %O_2_ for 1 week increased pulmonary NO availability, as marked by higher salivary NO and exhaled NO, but failed to improve hemodynamics, O_2_ saturation, and AMS development [132]. Animal models gave controversial results. On one hand, plateau pika (Ochotona curzoniae), a hypoxia-tolerant mammal that lives at 3000–5000 m/14.7–11.3 %O_2_ on the Qinghai-Tibet Plateau, show reduced NOx compared to lower-altitude controls, indicating that NO production is suppressed at altitude [133]. On the other hand, newborn highland lambs show enhanced NO function compared to lowland controls, corresponding to increased RhoA expression in the lungs, which suggests that NO-mediated vasodilation is important to maintain the pulmonary vascular resistance [134]. The general consensus, however, is that the NO level in lungs and plasma falls within 2 h following the hypoxia onset, after which it tends to return toward baseline levels by 48 h, followed by a further increase above baseline by 5 days [135]. Interestingly, melatonin, a pleiotropic hormone secreted in the pineal gland with antioxidant capacity that favors HIF-1α expression [136], depresses the hypoxia-induced increase in nNOS, eNOS, iNOS, and nitrotyrosine in hypoxic rats [137]. The estrogen receptors 1 and 2 seem to be associated with the activation of the NO–cGMP pathway during acute (7 days) acclimatization to 3500 m/13.8 %O_2_, being correlated to higher protein levels of eNOS, higher plasma NO_2_^−^ + NO_3_^−^, and lower levels of endogenous eNOS, an inhibitor of asymmetric dimethylarginine [138].

As for long-term hypoxia adaptation, a review of 32 articles published before 2012 revealed that Tibetans had the highest NO levels in the lung, plasma, and RBC of any highland populations [135]. Improved NO bioavailability thus appears to favor human adaptation to altitude and underlies the Tibetans’ response to altitude hypoxia [139]. This is compatible with both the increased plasma L-arginine observed in dwellers at 3800 m/13.3 %O_2_ for 10 months [140] and the beneficial effects of supplementing L-arginine [141] or sildenafil [142] at altitude. Remarkably, Amhara native highlanders in Ethiopia, who do not display erythrocytosis but have normal Hb–O_2_ saturation, display increased NO and cGMP, suggesting an adaptive pattern via vasodilation with cerebral circulation sensitive to NO but not to hypoxia [143]. In Ladakhi highlanders residing at 3500 m, circulating estrogen receptor 2 was higher than in sea-level controls with lower testosterone and higher eNOS protein level and plasma NO_2_^−^ + NO_3_^−^ [144].

In summary, NO handling appears to play an important role in hypoxia adaptation, with this feature particularly relevant for very long (generations) hypoxia exposure.

#### 3.2.2. Capillary Density

The Krogh capillary model is widely used to simulate the PO_2_ variation along a cylindrical capillary and to describe the O_2_ supply to the surrounding tissue [145]. The Krogh model helped understand the microcirculation in situations related to exercise and cerebrovascular dysfunction [146]. Despite its limitations, this model enables the appreciation that tissue oxygenation is regulated principally by capillary recruitment, with important effects of changes in the O_2_ gradient, i.e., hypoxia, in hematocrit and in the RBC transit time [147]. Because the hypoxia-induced reduction in the O_2_ gradient across the endothelium intuitively has a major role in determining O_2_ diffusion across the circulation-to-cell interface, and because the efficient flux of O_2_ from the circulation to the cell may represent a major bottleneck especially for high-O_2_ consumers such as muscle and brain, we will discuss separately these cases in terms of hypoxia adaptation.

As for the muscle, near-infrared spectroscopy enabled non-invasive assessment that the O_2_ diffusion in the muscle decreases with the hypoxia-induced fall of the O_2_ gradient, but this may be compensated by enhanced capillary density [148]. Therefore, augmenting the capillary density, or in the case of muscle the capillary/fiber ratio, or angio-adaptation, may help hypoxia adaptation through matching the O_2_ delivery to the varying metabolic needs of the myocytes [149]. In a pioneer study in guinea pigs native to the Andes, the capillary supply to skeletal muscle was 30% higher than in animals raised at sea level, highlighting that greater oxidative capacity necessitates increased capillarity, shorter diffusion distances for O_2_, and higher muscle myoglobin concentration [150]. Another study in Andean coot (*Fulica americana peruviana*) native to 4200 m/122.6 %O_2_ documented that, despite lower muscle fiber diameter, the number of capillaries per unit area and the capillary/fiber ratio were higher in all muscles compared with animals born at sea level [151]. Such data obtained in animals, perhaps adapted to altitude, however, were not confirmed in humans. Morphological studies in the skeletal muscle of humans exposed to 5000 m/11.3 %O_2_ for 8 weeks showed that the increase in the capillary/fiber ratio was not due to an increase in angiogenesis as for the maintained capillary network, but rather to the reduction in muscle cross-sectional area and muscle fiber size, both attributed to the loss of myofibrillar proteins [152]. The inadequate mammalian skeletal muscle angiogenesis observed after 8 weeks at 4500 m/12.0 %O_2_ was also addressed in biochemical studies that showed decreased markers of angiogenesis, e.g., vascular endothelial growth factor (VEGF), flt-1, and flk-1 mRNA [153]. The same trend was also observed in patients with chronic heart failure, where no appreciable changes in capillarity and fiber area were reported [154].

Theoretical analyses to assess whether the capillary network growth may represent an adaptive response to hypoxia in muscle predicted that increasing the capillary/fiber ratio above 2 may not be of benefit for tissue oxygenation [155]. Thus, capillary density might not appear as an adaptive response to hypoxia. This apparently counterintuitive feature could be justified by focusing on some limits of the Krogh model. Most important, the architecture of the microcirculation based on the Krogh cylinder may become insufficient to describe the O_2_ delivery to tissues because of the non-exclusive role of capillaries as suppliers of O_2_ to tissues. Indeed, the arterioles may represent a significant source of O_2_, as evident from data obtained with phosphorescence decay techniques for the measurement of local intra- and extravascular PO_2_ [156]. Findings on vascular longitudinal gradients converge in establishing that arterioles exert autocrine O_2_-driven effects in regulating the blood flow in the tissue [157], and that the blood viscosity, the blood O_2_-carrying properties, and the slope of the Hb–O_2_ dissociation curve are variables that should be accounted for to fully describe the O_2_ transport to tissues [158]. Additional variables such as local heterogeneities in RBC transit time and in hematocrit along the capillary have also been highlighted as critical to match the O_2_ availability to the varying metabolic demand [159]. As a matter of fact, a dramatically increased capillary density with reduced diffusion distance after short-term immobilization did not appreciably improve the O_2_ uptake in working skeletal muscle [146].

As for the brain, to defend its function in the face of O_2_ supply fluctuations, it triggers mechanisms that increase the capillary density and cerebral blood flow through vascular remodeling driven by HIF-1α overexpression with concomitant activation of downstream genes, especially VEGF [160]. Sustained hypoxia (28 days at 4500 m/12.0 %O_2_) causes remodeling of capillary vessels by increasing their diameter and length [161], and 28 days at 5900 m/10.0%O_2_ increases the vascular markers CD34 and PECAM-1 [57]. Increasing evidence points at O_2_-dependent regulation of the brain capillary network by neural stem cells that reside in niches within the brain, whose proliferation and multipotency is enhanced by mild hypoxia [162]. The role of HIF-1α in this mechanism is unclear, as in some studies it appears to be required [163], while in others it only facilitates the signal transduction pathways that promote self-renewal of neural stem cells with inhibition of apoptosis [164]. Alternative mechanisms may involve Wnt/β-catenin signaling [165], the calcium-regulated calcineurin-NFATc4 [166], and ROS [167].

After the initial discovery that hypoxia regulates VEGF gene expression in endothelial cells promoting mechanisms that impact the permeability and proliferation of the endothelium in an autocrine manner [168], a plethora of studies addressed the hypoxia-driven regulation of angiogenesis through an ever-expanding number of pathways in multiple cell types [169]. Today, hypoxia-derived VEGF overexpression is a globally recognized critical factor in malignant tumor expansion and vascular function [170], which is subjected to HIF regulation as an angiogenic master switch [171,172]. Indeed, of the several targets of HIF [49], the most involved here is VEGF along with several angiogenesis markers [173,174].

In summary, although changes in the capillary density do not appear pivotal in determining hypoxia adaptation in heavy-O_2_-consumer tissues such as muscle and brain, the scenario is different in tumor tissue, especially solid tumors, where the growth of O_2_-consuming malignant cells exceeds the growth of the capillary network, which finally decreases the capillary/cell ratio, inducing local hypoxia.

### 3.3. Arterial Blood-to-Capillary Interface

#### 3.3.1. Erythropoiesis

EPO, a glycoprotein synthesized in the kidney interstitial fibroblasts, has long been recognized to stimulate erythropoiesis in the bone marrow [175] in response to hypoxia [176]. The discovery of the underlying mechanisms was related to the discovery of HIF, as the effects of O_2_ deficiency on HIF’s stabilization and those of HIF on EPO production were the key to understand the pleiotropic mechanisms of the O_2_ sensors [177]. Once produced, EPO binds to the EPO receptors expressed in the cells of multiple tissues, including bone marrow and neurons, that activate JAK-2, which turns on the STAT, PI3K/Akt, and Erk pathways with multiple outcomes including erythropoiesis [178].

A plethora of studies converged in pointing out that EPO acutely responds to the onset of hypoxia within a few hours, peaks in the first 1–2 days, then progressively returns to baseline values in 1–2 weeks. The EPO peak is followed by a slower weeklong increase in the RBC mass [179]. However, despite the return of EPO to baseline values within a week after the onset of hypoxia, the erythropoietic stimulus persists for at least 10 months at altitude [180]. An important player in this mechanism, the soluble form of the EPO receptor, an endogenous antagonist of EPO, was observed to mirror the EPO response to hypoxia, decreasing by 19%, remaining below baseline values for at least 72 h at 4340 m/12.4 %O_2_, and then slowly returning to baseline values [181]. This highlights the EPO-to-EPO receptor ratio as the key to understand the long-term relationship between EPO and the RBC mass.

Further insight into the mechanism underlying the path hypoxia→HIF→EPO→erythropoiesis comes from the comparative analysis of permanent altitude dwellers. Tibetans do not have marked erythrocytosis [182]. Accordingly, high- and low-living Tibetans display the same EPO levels [183]. Furthermore, the EPO level is low in Ladakhi highlanders, who are ethnically linked to Tibetans [184]. Likewise, Ethiopian Amharas, who live at comparable altitudes, have no erythrocytosis compared to their sea level counterparts [185]. By contrast, Andean altitude dwellers are well-known to have excessive erythrocytosis [186], which in most cases foreplays the onset of CMS. One would expect that their EPO level would be higher, but this is not the case. Indeed, in South American dwellers at 4340 m/12.0 %O_2_, circulating EPO does not appear as a discriminating factor for CMS, but the EPO-to-EPO receptor ratio may be the factor that determines excessive erythropoiesis [187]. Likewise, the EPO level was only marginally higher in Andean highlanders with high and low Hb [188].

Augmenting the production of RBCs is perhaps the first physiological response ever recognized to participate in hypoxia adaptation [189]. This response, which was later demonstrated to be mediated by the hyperproduction of EPO [190], was considered adaptive because it translates into a proportionally higher O_2_ carriage to tissues [191]. The main reason for the injurious effects of polycythemia on blood O_2_ transport resides in the increased shear rate due to high blood viscosity, which critically determines local blood flow [192]. Along with factors such as fibrinogen, RBC deformability, and aggregation, the hematocrit is indeed the single most important determinant of blood viscosity [193]. In healthy humans, hematocrit variation within the normal 33–45% range results in modest alterations in cerebral blood flow, but beyond this range, hematocrit correlates with decreased blood flow [194].

In summary, the observation that adapted populations exhibit a blunted erythropoietic response, in contrast with the exaggerated response often associated with the adverse clinical features observed in non-adapted populations, suggests that erythropoiesis might not be an adaptative response to hypoxia.

#### 3.3.2. Iron Handling

Augmented erythropoiesis inevitably leads to iron stress. As shown in Figure 7, Hb synthesis implies the intervention of at least two factors: the stimulus driven by hypoxia and EPO discussed above, and the supply of iron, which must thus be linked to the intensity/duration of hypoxia [195]. In physiological hypoxia, the greater need for iron to build up Hb is met through increased absorption from the gut or from the liver, which acts as an iron storage organ. The main iron transporter across the basolateral membrane of enterocytes and hepatocytes, ferroportin, is under tight regulation by hepcidin, which internalizes ferroportin, decreasing the capacity to export iron into the circulation [196,197]. During acute hypoxia (days–weeks), hypoxia downregulates hepcidin, thereby favoring the entry of iron into the bloodstream and hence erythropoiesis [198].

In pathologic hypoxia, however, another scenario occurs. COPD is a disease in which airflow obstruction causes a significant decrease in endothelial function over time and hypoxia [199]. While 40–50% of COPD patients undergo physiological erythropoietic adaptation to hypoxia, 5–30% of them instead develop iron deficiency with anemia [200,201]. Remarkably, anemia surges as a predictive risk factor for worse outcomes [200,202,203]. In ARDS, which is often characterized by severe hypoxia [204], the development of anemia in patients represents a recognized comorbidity. This divergent response to physiological and pathological hypoxia has been attributed to different regulation of hepcidin driven by the redox imbalance, which is higher in ARDS and COPD patients and represents an additional factor that regulates erythropoiesis in the presence of strong inflammatory stimuli, such as the level of C-reactive protein [205]. Remarkably, patients affected by COVID-19, especially the first waves, were often characterized by severe hypoxia as in ARDS patients. It is debated whether COVID-19 resembles ARDS in respiratory failure development, lung compliance, and endothelial inflammation [206], but the redox imbalance and the cytokine storm seem to be even higher in COVID-19 than in ARDS [205]. Sphingolipid synthesis is selectively stimulated by iron-driven inflammation [207], especially sphingosine-1-phosphate, a mediator in COVID-19 syndrome [208].

In summary, the iron-handling response to physiological hypoxia, at least in the absence of inflammatory factors, increases erythropoiesis, which may not be seen as a favorable factor. Therefore, in physiological hypoxia the downregulation of hepcidin may be interpreted as maladaptive.

#### 3.3.3. Hemoglobin Oxygen Affinity

The O_2_-carrying capacity of blood depends mainly on the blood Hb concentration (discussed elsewhere) and the Hb–O_2_ affinity, which determines the Hb–O_2_ saturation at any given PO_2_ as a function of pH, PCO_2_, the intraerythrocytic level of 2,3-diphosphoglycerate (DPG, a glycolytic intermediate that binds to the T-form of Hb and acts as an effector of the Hb function), temperature, and the level of carbon-monoxide-bound Hb [209]. The P50, i.e., the PO_2_ at 50% Hb–O_2_ saturation, is a useful index of the Hb–O_2_ affinity. Diminished Hb–O_2_ affinity, or higher P50, is believed to be favorable at altitudes ranging from sea level to moderate because it augments the arteriovenous O_2_ difference by decreasing the venous Hb–O_2_ saturation [210]. From Fick’s equation, which relates VO_2_ to the cardiac output and the arteriovenous O_2_ content difference, it can be argued that a lower Hb–O_2_ affinity or higher P50 may translate either to a better tissue oxygenation (venous PO_2_ = +3.2 mmHg for a P50 change of 1 mmHg) or to a lower circulatory load (cardiac output = −5.8% for a P50 change of 1 mmHg). By contrast, at extreme altitudes, the O_2_ delivery to tissues is favored by a high Hb–O_2_ affinity, or lower P50, because it compensates the compromised arterial Hb saturation [211]. An arterial pH shift of 0.1 units increases the Hb–O_2_ saturation by up to 5% at 6450 m [212]. It is thus expected that, at moderate altitude, alkalosis, which increases the Hb–O_2_ affinity and whose genesis will be discussed below, is deleterious because it lowers the P50, while increased DPG is instead favorable for the opposite reason. As shown in Table 2, the changes in the two factors that were measured up to 6450 m altitude appear to offset each other almost perfectly, leaving the P50 value nearly unchanged, in agreement with early hypotheses [213].

It should, however, be highlighted that two seldom-analyzed factors must be accounted for a full appreciation of the relevance of the blood O_2_ transport picture at altitude. First, the fluctuations in temperature, which may be non-trivial especially in extreme environments: a change of 1 °C shifts the P50 by about 1.5 mmHg [214], a relevant value with respect to those indicated in Table 2. Second, the altitude-induced erythropoietic stimulus admits in the circulation an increasing fraction of newly synthesized, “young” RBCs [215]. When RBCs emerge from the bone marrow, they have a higher DPG/Hb ratio than the “old” ones (0.96 ± 0.13 vs. 0.57 ± 0.13 mole/mole) [216], which corresponds to a P50 change from 29.1 to 24.8, i.e., a 4.3 mmHg change, which is much higher than those reported in Table 2. Thus, in circulating blood with a normal average DPG/Hb, the RBC distribution in terms of O_2_ affinity may be very wide. Unfortunately, it is not possible to predict which type of RBC actively participates in the O_2_ exchange in the capillary or is preferentially diverted to the arteriovenous shunt. However, if flexible RBCs are predicted to undergo the tortuous path along the capillary, then it is likely that “young” RBCs with a lower mean corpuscular Hb concentration and lower O_2_ affinity because of their higher DPG/Hb ratio [215] are those that actively release O_2_ with respect to old, sticky (due to their higher mean corpuscular Hb concentration), RBCs with a low DPG/Hb ratio and high O_2_ affinity. We recognize that the lack of any data in this respect may render this discussion highly speculative.

In summary, although the changes in Hb–O_2_ affinity may in principle participate in hypoxia adaptation, it appears that they, if any, are very small and may not be considered pivotal to postulate a relevant role during hypoxia adaptation.

### 3.4. The Alveoli-to-Arterial Blood Interface and the Circulatory Responses

#### 3.4.1. Cardiac Output

The responses to acute hypoxia encompass increased cardiac output in an attempt to deliver more O_2_ to tissues by a mechanism that occurs mainly through a compensatory increase in heart rate [217]. At the base of this response, the adrenergic G protein system emerges, as evident from elevated levels of catecholamines and adrenergic fiber activity [218]. The role of stroke volume alterations in modifying high-altitude changes in cardiac output is controversial, as most studies did not observe relevant changes in stroke volume after acute exposure to hypoxia [219]. Another mechanism behind cardiac output, the reduction in systemic vascular resistance, was recently observed in healthy individuals in hypoxic chambers simulating high-altitude exposure [220]. Peripheral arteriole dilatation [221] further helps matching the tissue O_2_ demand/supply ratio [222]. Despite such a mechanism, a decreased stroke volume at high altitude was observed in several studies. Impaired systolic contractile function secondary to reduced coronary PO_2_ has been investigated as a potential cause [223]. At the same time, as the basal cardiac output is increased, there is less chance for an exercise-related cardiac output increase in response to exercise [224].

The cardiac output, on the other hand, is not as much modified by persistent hypoxia. In the long term, the β-adrenergic receptors are downregulated [225], while muscarinic receptors are upregulated with alterations in the expression and activity of both inhibitory and stimulatory G proteins and a decreased response of the adenylate cyclase system [218]. Cardiac chronotropic function could be controlled by a local mechanism linked to myocardial PO_2_ [226], which leads to a decrease in compensatory tachycardia in order to reduce VO_2_ further.

In summary, the acutely observed changes in cardiac output, combining heart rate and stroke volume, do not appear to sustain long-term hypoxia and hypoxia adaptation.

#### 3.4.2. The Red Blood Cell Pulmonary Capillary Transit Time

RBCs might not have enough time in the capillary to uptake all O_2_. The binding of Hb with O_2_ has a finite reaction rate, but the rate of this reaction, which is accomplished in <10 msec [227], is much faster than the diffusion rate. The limiting factors that influence the O_2_ diffusion in the lung, after assuming irrelevant impacts of the O_2_ diffusion coefficient, temperature, fluid viscosity, chemical properties of the membranes, and surface area of the pulmonary gas exchange, thus are (a) the alveolar-to-arterial PO_2_ difference, (b) the blood–gas barrier thickness, and (c) the RBC capillary transit time. Both acute and chronic hypoxia may have substantial effects on these factors.

The alveolar-to-arterial PO_2_ difference is assessed through the ventilation/perfusion (V/Q) ratio, which is known to be impaired by hypoxia, especially in acute situations. Indeed, relative hypoventilation mismatches V/Q, but this is compensated by pulmonary vasoconstriction that redirects the blood flow from poorly ventilated areas to well-ventilated areas in the lung, resulting in better oxygenation. This phenomenon is understood to be the primary active regulator of V/Q matching. A recently developed computational model summarizes ongoing research in this field, as it predicts that hypoxic pulmonary vasoconstriction matches perfusion to ventilation, homogenizes regional alveolar–capillary O_2_ flux, and increases O_2_ loading in the circulation by improving V/Q [228].

The thickness of the basal lamina, a layer of extracellular matrix formed by epithelial cells that acts as an attachment point for cells, may constitute a major resistance element in the pulmonary blood–gas barrier, representing a compromise between two needs: to provide mechanical resistance against excessive pressure and to facilitate gas diffusion across the alveolar barrier [229]. This compromise may break in pulmonary pathologies linked to hypoxia-induced pulmonary edema due to stress failure of pulmonary capillaries in PH [230] as well as in COPD [231]. Experimental data obtained in rats exposed to 5900 m/10.0 %O_2_ for 2 weeks show that the thickness of the basal lamina nearly doubles, but this increase is efficiently prevented by phosphodiesterase 5 inhibitors in parallel with other parameters highlighting the development of PH and right-heart failure [232]. Indeed, the HIF-1α pathway has been shown to be activated in alveolar type II cells after lung injury, promoting proliferation and spreading during repair [233].

A higher cardiac output in acute hypoxia translates into proportionally shorter RBC transit times. In normal conditions, the average transit time of RBCs in the alveolar capillaries is ∼0.75 s, despite a very wide distribution of transit times in the lungs [234]. In excised rabbit lungs, no limitation to oxygenation were observed for a sixfold increase in blood flow [235]. The competition between shortening RBC transit time and full RBC equilibration with alveolar gasses was examined [236] with the Fick diffusion equation in a simple boundary analysis to estimate the external resistance to O_2_ diffusion for RBCs [237]. While the rate of the Hb–O_2_ loading is not a critical parameter at sea level because of the steep alveolar-to-arterial PO_2_ difference, this may not hold true at altitude, where low alveolar O_2_ limits O_2_ diffusion and may impair the equilibration of RBCs with alveolar O_2_. In fact, O_2_-demanding conditions such as working at 3840 m/13.2 %O_2_ imply a lack of full equilibration due, at least in part, to shortened RBC transit times [238].

Sustained hypoxia is expected to determine pulmonary remodeling, the basis for the onset of PH [239]. This process involves the mobilization of bone-marrow-derived progenitor cells that express the transmembrane tyrosine kinase receptor c-kit [240,241]. In a model of chronic 2-week hypoxia at 10 %O_2_, rat lungs exhibited a tenfold increase in c-kit-positive cells, accompanied by marked vessel muscularization and redistribution of pulmonary capillary size, with a great increase in small (<50 μm) capillaries vs. an unchanged pattern of larger capillaries [242]. This pattern was observed nearly unaltered for double hypoxia times, indicating that it might be part of hypoxia adaptation [243]. However, it is not yet possible to predict the impact of varying pulmonary capillary size on the RBC transit time and the role of the alveoli-to-arterial blood interface on hypoxia adaptation.

What was discussed above for the RBC O_2_ loading in the lungs may also apply in reverse to the O_2_-unloading process in the peripheral tissues but is complicated by the relatively slow rate with respect to O_2_ binding by two orders of magnitude [237]. This might render the RBC O_2_ unloading problematic in the case of short transit times, low PO_2_ gradients across the endothelium, altered viscosity due to excessive erythropoiesis, Hb–O_2_ affinity changes, modifications in the acid–base status, and other factors. However, few data are presently available on this topic. However, an analysis showed that the diffusing O_2_ does not encounter resistance in the passage through the RBC membrane but rather in the unstirred layer on the RBC surface [237].

In summary, the RBC pulmonary capillary transit time, and perhaps to a minor degree the peripheral capillary transit time, may represent two important features that determine the response to acute and chronic hypoxia. However, more detailed studies on the regenerative capacity of the pulmonary capillary network are needed to understand the relevance of this phenomenon in terms of hypoxia adaptation.

### 3.5. The Atmosphere-to-Alveoli Interface and the Ventilatory Responses

#### 3.5.1. Carotid Bodies

With progressive hypoxia, the ventilatory rate augments in an attempt to favor blood oxygenation in the alveoli. The carotid body (CB), a cluster of cells located in the adventitia at the fork of the common carotid artery, is the main chemoreceptor that senses the insufficiency of O_2_ and turns on this reflex [244]. A series of mechanisms enables the CB glomus type I cells to perform the O_2_-sensing function. A major one involves to the inhibition of the O_2_-sensitive K^+^ channels [245], which leads to cell depolarization, increased Ca^++^ entry, and the release of intracellular vesicles containing neurotransmitters such as dopamine, acetylcholine, ATP, and catecholamines [246] that increase the ventilatory rate [247]. This mechanism requires a functional electron transport chain with intact coupling between mitochondrial complexes I and IV [248]. The downregulation of mitochondrial complex IV activity increases the production of NADH and ROS, which negatively modulate the membrane ion channel activity [249,250].

Alternative mechanisms may affect the O_2_ sensing by the CB glomus cells as well. First is the release of gaseous messengers not stored in vesicles such as NO, carbon monoxide, and hydrogen sulfide that antagonize the response to hypoxia by inhibiting CB excitation with depression of the ventilatory responses [251]. By contrast, EPO at low concentrations (<0.5 IU/mL) agonizes the CB response to hypoxia [252]. Indeed, EPO is emerging as an effective prophylactic treatment not only for the treatment of AMS, but also for a variety of acute respiratory and non-respiratory symptoms of SARS-CoV-2 infection [253].

Clearly, the mechanisms underlying the O_2_-sensing activity in the CB still need more insight to justify a few recent observations. For example, there is an attenuated CB sensitivity to hypoxia in naked mole rats, which exhibit a blunted hypoxic ventilatory response despite larger CBs with more glomus cells than in mice, perhaps due to the interruption of gas-messenger signaling by heme oxygenase-1 [254]. In addition, high-altitude deer mice display attenuated ventilatory sensitivity and CB growth, perhaps associated with a genetic variation in HIF-2α [255]. Finally, the transient receptor potential ankyrin 1 is a cation channel broadly expressed in the trigeminal and vagus nerves that can detect irritants in inspired gasses and is activated by mild (4000–2500 m/13.0–15.0 %O_2_) but not severe (5900 m/10.0 %O_2_) hypoxia [256]. Intriguingly, SARS-CoV-2 CB infection could be the cause of the “silent hypoxemia” observed in COVID-19 patients, which may warrant the use of CB activators as respiratory stimulants in COVID-19 patients [257]. The mechanisms illustrated for the CB are shared in principle by other O_2_ sensors such as the aortic bodies and astrocytes [258].

In summary, the CB response to hypoxia is a recognized pattern with a potentially strong impact. While moderate increases in the ventilation rate might appear protective and useful to compensate the effects of hypoxia, strong increases may by contrast appear maladaptive, such as for the risk to cause hyperventilation-driven alkalosis, as discussed in the next sub-chapter.

#### 3.5.2. Hyperventilation and Alkalosis

As a consequence of hypoxia-driven hyperventilation, arterial PCO_2_ decreases due to excessive CO_2_ washout from the lungs, causing alkalemia. Alkalosis is in part compensated by the kidney through a metabolic adjustment that establishes a negative base excess. Different patterns were observed at 6450 m/9.3 %O_2_ in unacclimatized Caucasians and better-acclimatized Sherpas [212]. It is hard to believe that Sherpas may ever adapt to that altitude despite their long-time, perhaps millennia-long [259], exposure to 4000–5000 m/12.9–11.3 %O_2_, but their degree of acclimatization to 6450 m/9.3 %O_2_ is almost surely better than that of Caucasians independently of the training degree. Despite the same arterial PO_2_ and O_2_ saturation (Figure 8), arterial PCO_2_ is lower in Caucasians than in Sherpas, perhaps due to a blunted respiratory sensitivity to hypoxia in Sherpas. An irreversible blunted respiratory drive was observed in high-altitude natives [260] and was confirmed in preterm infants at altitude [261] and in obesity hypoventilation syndrome [262] and has been linked to the development of the so-called “happy hypoxemia” [263]. Blunting the respiratory drive appears protective in Sherpas exposed at extreme altitudes because it depresses the need for excessive ventilation, which reduces alkalosis. Remarkably, the base excess was −6 mEq/L in both Caucasians and Sherpas, compared to the −10 mEq/L that would have elicited full compensation of alkalosis, highlighting that the metabolic compensation operates to the same extent for both. However, depressed ventilation in Sherpas compared to Caucasians produced a lower arterial pH. The apparently small changes in the pH values should not be misleading: while at pH 7.4 [H^+^] = 3.98 × 10^−8^, at pH 7.45 and 7.5 [H^+^] = 3.54 × 10^−8^ (−10%) and [H^+^] = 3.16 × 10^−8^ (−20%) of the initial value, respectively, which is a remarkable decrease. AMS sufferers displayed even greater alkalosis: pH 7.57–7.63, or [H^+^] = 2.51 × 10^−8^, i.e., −37% of the normal value.

Thus, correcting hypoxemia through hyperventilation might represent a costly option because of the onset of alkalosis. The effect of persistent alkalosis on brain function has not yet been dissected in detail, but it may pave the road to at least two undesirable events associated with high plasma pH: (a) promoted binding of Ca^++^ to albumin, which reduces blood ionized Ca^++^ at constant total calcium levels and keeps neurons and muscles in an irritable state (tetany); and (b) vasoconstriction of the brain’s blood vessels [265], which impairs brain function. Direct data on cognitive impairment at altitude is scarce but two meta-analyses revealed negative effects of hypoxia on cognition [266,267], a finding further confirmed in lowlanders exposed for 3 days at 3269 m [268]. The effect of hypoxia on cognitive function in humans residing at altitude is now recognized [269,270,271,272], and O_2_ supplementation in high-altitude schools has been recommended to prevent the impairment of learning processes [24].

In summary, hyperventilation appears as an excessive response to hypoxia in consideration of CO_2_ depletion with alkalosis, and may conflict with wellness, which fits the definition of maladaptation.

## 4. Adaptation vs. Maladaptation

### 4.1. Ability of Sea Level Populations to Adapt to High Altitude

Most literature data are obtained in the days or weeks immediately following the onset of hypoxia, where the signs of maladaptation are addressed in terms of AMS insurgence. Relatively little data are available to document the effects of longer (months, years) exposures, which may give clues for the ability of low-altitude dwellers to adapt to hypoxia.

The sojourners in the Antarctica plateau are fit and healthy individuals who are specifically trained to sojourn for up to 10 months at an equivalent altitude of 3800 m/13.2 %O_2_ in an environment that excludes the presence of disturbing variables related to cold and changes in altitude, with the only probable exception the rupture of circadian rhythms [180]. In these subjects, the expected changes in the acid–base status are maintained for 10 months without any appreciable modification [180]. Remarkably, even the metabolome changes remain unaffected for that period of time, with the exception of a slow return toward baseline of the non-polar metabolome after 6 months of hypoxia [140]. Likewise, the variables linked to the redox imbalance, including ROS, oxidative stress biomarkers, NO, and proinflammatory cytokines, peak by the 20th day of hypoxia but do not return to baseline levels [67]. Such data highlight a missed capacity of adaptation to hypoxia in Caucasians born at sea level, at least for one year. Longer permanence of sea-level dwellers at altitude, such as immigrant Han Chinese in Tibet, gives rise to up an to 18% incidence of CMS [273]. Thus, to the best of our knowledge, available data indicate that sea-level-dwellers exposed to hypoxia for years have missed the ability to adapt to this challenge.

### 4.2. Generations-Long Adaptation to Hypoxia

About 80 million people live permanently at >2500 m altitude in the Andes, South-East Asia, and Ethiopia [1]. These populations have developed diverging responses to altitude hypoxia. Andeans go the hematological route [274] with elevated Hb that enables carrying more O_2_ in blood, but high CMS incidence, which occurs in 16% of the adult male population, with increasing prevalence with age, rising up to 30% by the fifth decade of age [275], highlighting poor adaptation to hypoxia. Tibetans go the respiratory route [274]: they inhale more air with each breath and breathe more rapidly with high plasma NO_2_^−^ + NO_3_^−^, a biomarker of the NO-storing capacity. This results in sporadic CMS in altitude-native Tibetans [273,276]. Ethiopians Amhara highlanders display lower CMS rates [277], reduced erythropoiesis, higher plasma NO_2_^−^ + NO_3_^−^ and cGMP, and lower diastolic blood pressure. These features translate into a marked vasodilatory response to hypoxia with respect to related lowlanders at altitude, who instead display an elevated erythropoietic response [278]. Although not properly an altitude population, the Kyrghyz commuters are of interest because 14–20% of them show signs of altitude PH and are characterized by a higher-than-normal fraction of hyper-responders to acute hypoxia [279]. A case–control study identified in healthy Kyrghyz highlanders the presence of genetic traits that discriminate hyper-responders to hypoxia who develop PH [280]. It was pointed out [280] that Kyrghyz commuters have unique patterns with respect to other altitude populations because they do not present other features linked to chronic hypoxia besides PH, unlike the Andean population, where polycythemia may have confounding characteristics, or Tibetans, where PH is practically absent.

It is therefore tempting to state that high-altitude populations born and residing in the Himalayas (Tibetans, Sherpas, and to a lesser extent Ladakhis) are “adapted” to altitude because they suffer altitude-related diseases to a lesser extent and display a reduced erythropoietic response. By contrast, Andean populations (Aymaras and Quechuas) reveal an exaggerated erythropoietic response with hematocrit values exceeding 50% and an increased risk of dysfunction due to high blood viscosity [259,281,282]. A hypothesis explaining the Tibetan–Andean differences is the longer altitude residence for Tibetans (about 30,000 years), with enough time to adapt genetically, than for Andeans (about 10,000 years) [283]. These figures are, however, complicated by the relatively recent massive migration into the Andes in the XVIIth century, as opposed to the relatively conserved population distribution in the Himalayas.

### 4.3. Ability of High-Altitude Populations to Adapt to Sea Level

Although uncommon, in some instances altitude-adapted subjects are forced, mostly for political reasons, to flee their native habitats in search of more suitable environments, often at lower altitudes. Several thousand Tibetan refugees born at altitude but residing at sea level provide the unique opportunity to test the reversibility of the processes that may have driven altitude adaptation. In this instance, the hypoxic stimulus is broken, and the subjects are exposed to a condition of relative hyperoxia. Figure 9 (Samaja et al., unpublished observations) shows the blood Hb concentration in individuals born in Tibet who have fled to <500 m/>20.1 %O_2_, and in Indian-ancestry residents in the same area as a control. Clearly, missing the hypoxic stimulus remarkably depresses erythropoiesis, at least in males, who face the risk of becoming anemic. Another large-scale study revealed a lower RBC count, hematocrit, and Hb levels in Tibetans living long-term at low altitude compared to their high-altitude counterparts [183]. In a further study, it was found that the Hb concentration was lower in Tibetans living at sea level than in Han Chinese individuals, along with a higher minute ventilation and blunted pulmonary vascular responses to acute (minutes) and sustained (8 h) hypoxia [284]. The same study also shows a lower hypoxic induction of HIF-regulated genes in peripheral blood lymphocytes as well as a significant correlation between EPAS1 and EGLN1 genotypes and induction of EPO by hypoxia in Tibetans compared with Han Chinese, highlighting the less-vigorous response to hypoxic challenge [284]. This evidence converges in indicating the occurrence of outcomes compatible with a response to a relatively hyperoxic environment in subjects adapted to live at altitude. It remains to be established if such a hematological response to relative hyperoxia is harmful.

## 5. A Glance to the Future: The -Omics

### 5.1. Genomics

Marked discordant responses to altitude provide a unique opportunity for genome-wide association studies (GWAS) to identify the gene patterns that determine different responses to hypoxia across the world. More than 1000 genes appear to be potentially involved in the response to hypoxia in the three altitude populations [285]. These genes are related to the development of the nervous, lymphoid, and cardiovascular systems, as well as with cell growth, survival, and death, with only four genes (PIK3CB, HLA-DQB1, CNTNAP2, and DLG2) commonly altered in all the populations [285]. Such findings highlight that hypoxia adaptation is polygenic and impacts many biochemical paths, each of which in turn affects many physiological responses. Despite substantial convergence on a few physiological mechanisms (RBC homeostasis, O_2_ transport, circulatory adjustments, and O_2_ sensing) as key paths that affect hypoxia adaptation, other GWAS could not distinguish the roles played by certain genes believed to be associated with excessive erythropoiesis [286]. This emphasizes a major limitation of GWAS, i.e., the presence in the analyzed populations of confounding factors such as population migrations, geographical differences, and diet restrictions that may introduce disturbing signals not related to hypoxia.

In a powerful GWAS performed on >3000 Tibetans and >7000 non-Tibetan individuals of Eastern Asian ancestry, signals of hypoxia adaptation were detected at nine genomic loci, one of which corresponds to EPAS1 [287]. Another GWAS of high-altitude populations identified the O_2_-sensing machinery as responsible for the phenotypic adjustments leading to adaptation [288]. EGLN1 and EPAS1 play an important role in determining the variability in the response to hypobaric hypoxia at altitude and have been linked to the development of polycythemia and associated abnormalities [289]. Overall, it seems that when gene expression overresponds to the hypoxic challenge, maladaptive patterns occur. For example, exaggerated stimulation of the O_2_-sensing system leads to HIF overexpression, which in turn leads to excessive erythropoiesis and CMS-related pathology. Remarkably, suppression of HIF-2α in North American deer mice exposed to chronic hypoxia (4 weeks at arterial PO_2_ = 90 mmHg, or 4630 m) attenuates carotid body growth and glomus cell hyperplasia without effects on hematology [255].

### 5.2. Metabolomics and Amino Acids

Amino acids are not only simple building blocks for proteins but may also have relevant functions as carriers of signals. Despite the scarcity of studies correlating hypoxia with amino acids, they may be highly involved in the responses to hypoxia and perhaps adaptation. Undoubtedly, the availability of -omics techniques may enable us in the future to take a deeper look at the potential role played by amino acids during hypoxia adaptation.

A metabolomic study performed in healthy lowlanders exposed for up to 10 months to 3800 m/13.2 %O_2_ altitude revealed a modest decrease in major lipids, but also marked alterations in amino acid metabolism, with increases in plasma arginine, glutamine, phenylalanine, tryptophan, and tyrosine [140]. Similar increases in plasma tryptophan and serotonin were also found in acutely hypoxic (19 days up to 6885 m/8.8 %O_2_) subjects [290]. Much work demonstrates higher plasma levels of some, but not all amino acids in the plasma of humans exposed to prolonged physiological and pathological hypoxia. This may be attributed to hypoxia-linked protein catabolism [291], but as the pool of proteogenic amino acids is not altered by hypoxia nor varied with time [140], it suggests insufficient hypoxia-induced protein catabolism, and the above changes in plasma amino acids indicate the occurrence of pathways that involve amino acids not as building blocks of proteins but as messengers.

Among the paths involving amino acids, the glutamine/glutamate balance is relevant for its implications in regulating the brain cognitive function and the ventilatory rate. Indeed, low glutamate stimulates ventilation to compensate lower O_2_ saturation [292], and high glutamine may affect the brain cognitive function by disrupting the ammonia vs. glutamate balance [293]. Such a disruption with decreased N-acetylputrescine, spermine, and ornithine also affects angiogenesis and reproductive physiology [294].

High levels of plasma tryptophan may reflect an impaired activity of tryptophan hydroxylase, which requires O_2_ and is the committing step in the synthesis of serotonin [295]. Downregulation of brain serotonin results in depression, reduced appetite, and disruption of sleep patterns, as commonly found in altitude residents [296,297]. Altered metabolisms of glutamate, threonine, taurine, lysine, arginine, and proline are frequently observed in pathological hypoxia such as in ARDS [298] and asthma patients [299]. Upregulation of amino-acid-related pathways was also observed in experimental perinatal asphyxia and neonatal hypoxia-ischemia encephalopathy [300], while in fetal growth restriction pregnancies the kynurenine pathway is enhanced [301]. This pathway is relevant because the linked nicotinamide pathway promotes pulmonary vascular remodeling [302], a common finding in chronic hypoxia [303]. Nicotinamide [304] and niacin [305] are being investigated as drugs to ameliorate the progression of hypoxia-induced pulmonary vascular remodeling.

### 5.3. Limitations of the Study

We omitted the issues related to biological systems in which the adaptation process does not yet emerge with clarity such as, for example, the immunological system and the microbiome. The rationale underlying the references to tumor tissue as a responder to pathological hypoxia is based on the paradigm that tumors may represent an example of successful adaptation to hypoxia, which translates into better viability and growth of tumor cells with respect to healthy cells. However, we avoided the extensive use of this paradigm in this study because this possibility has not yet been fully demonstrated and may appear speculative.

Several geographically isolated high-altitude mammals such as yaks [306,307], llamas [308], pikas [309], and many others have been considered models for the evolutionary adaptation to hypoxia. Comparison studies using such models may indeed constitute an appreciable experimental bench to test the credibility and suitability of the adaptation patterns observed in humans. However, each animal displays species-specific characteristics that would deserve separate analyses for each animal type and each genetic trait to enable meaningful comparisons with the layouts observed in humans. Therefore, despite being aware of this limitation, in this review we opted to focus on data obtained in human populations, as well as in controlled in vitro and in vivo laboratory contexts, thereby excluding valuable studies conducted on altitude-dwelling animals, especially for -omics studies.

We regret neglecting many precious contributions for reasons of space.

## 6. Conclusions

In the past eras, %O_2_ has fluctuated within the 0–30% range [310]. Although various forms of life could adapt successfully from time to time to the varying O_2_ level, it remains uncertain whether humans have lost the ability to adapt to lower-than-normal %O_2_. The analysis of the bottlenecks during the cascade of O_2_ from the atmosphere to the mitochondria might address the sites where the human organism succeeds or fails to adapt to hypoxia. Remarkably, many of the analyzed sites are practically superimposable to what was anticipated in an article published 70 years ago [311]. Extreme variability in targets, methods of analysis, as well as the presence of disturbing factors in the analyzed individuals or populations might have blurred to a considerable degree the judgement on how the responses at the sites along the O_2_ cascade concur in building up hypoxia adaptation. Some of the responses appear compatible with progressive adaptation, for example, the O_2_-sensing mechanism, the HIFs, the buildup of antioxidant defenses, the bioenergetic adjustment, and NO-mediated responses. By contrast, exaggerated CB sensitivity to O_2_ changes, with excessive hyperventilation, uncontrolled erythropoiesis, and iron handling, might appear as signs of maladaptation. Other responses, such as the changes in cardiac output and Hb–O_2_ affinity, appear irrelevant in terms of hypoxia adaptation. The observation that adaptation to hypoxia seems to be accomplished only after thousands of years of continuous hypoxia indicates that humans are animals not particularly suitable for living at altitude. As has already been pointed out [312], this should readdress high-altitude research, now based upon lowlanders from Western, educated, industrialized, and rich countries ascending to high altitude, toward understanding to a greater extent the principles of genes, cells, and molecular physiology in permanent high-altitude residents.

## Figures and Tables

**Figure 1 ijms-24-03670-f001:**
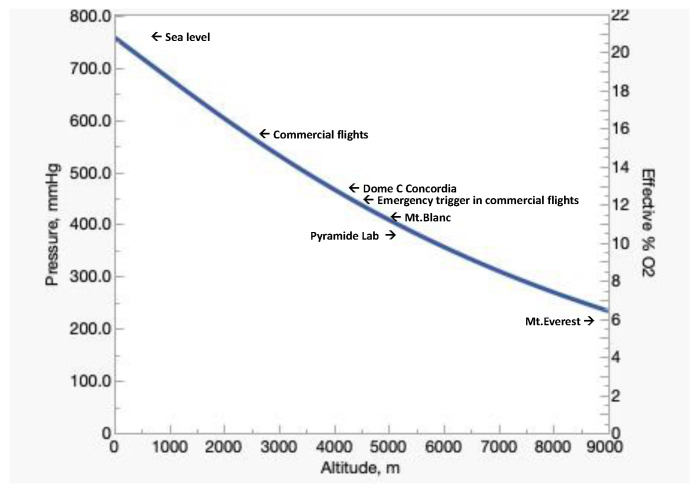
Behavior of the barometric pressure (left Y-axis) and of the effective %O_2_ (right Y-axis) at varying altitude above sea level in the Earth atmosphere. Data from the International Standard Atmosphere (ISA). The ISA does not contain water.

**Figure 2 ijms-24-03670-f002:**
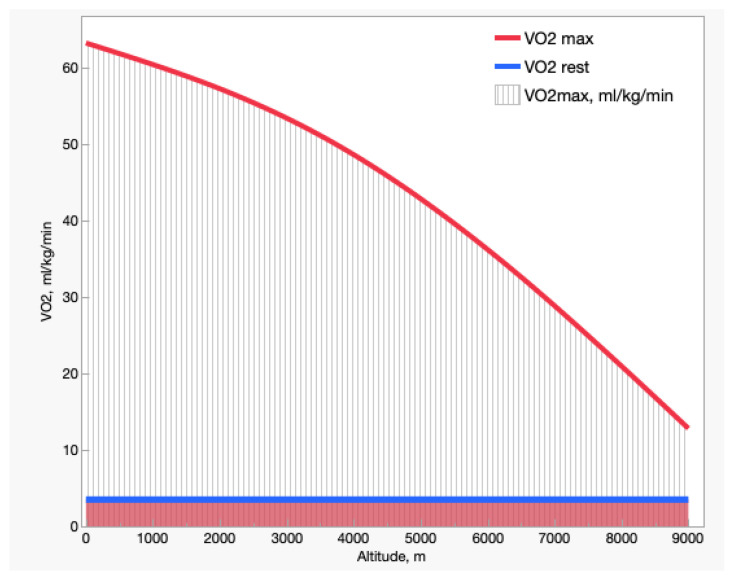
Behavior of maximal (red) and resting (blue) VO_2_ at altitude. Data redrawn from [12], which reports the average of 146 studies performed at various real and simulated altitudes in the 0–9000 m range regardless of the training degree. The dashed area represents the O_2_ available to perform exercise.

**Figure 3 ijms-24-03670-f003:**
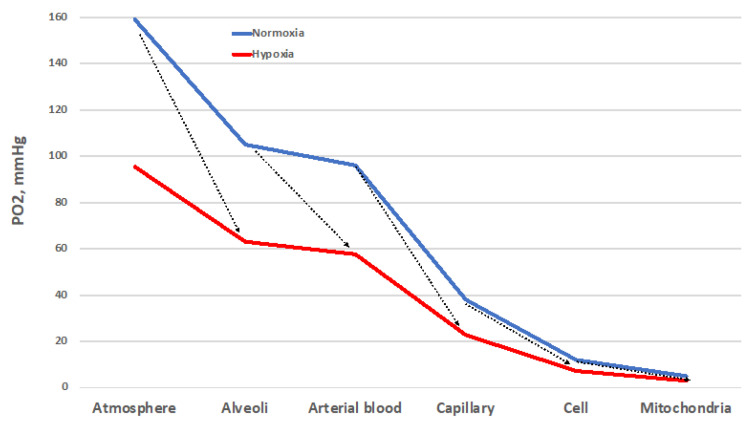
The oxygen cascade in normoxia (blue) and in hypoxia (red, which depicts the situation at 3400 m/13.9 %O_2_). The arrows indicate the sites of resistance to the O_2_ flux that increase the PO_2_ gradient. These sites of resistance may convert a normoxic situation into a hypoxic one. Other explanations in the text.

**Figure 4 ijms-24-03670-f004:**
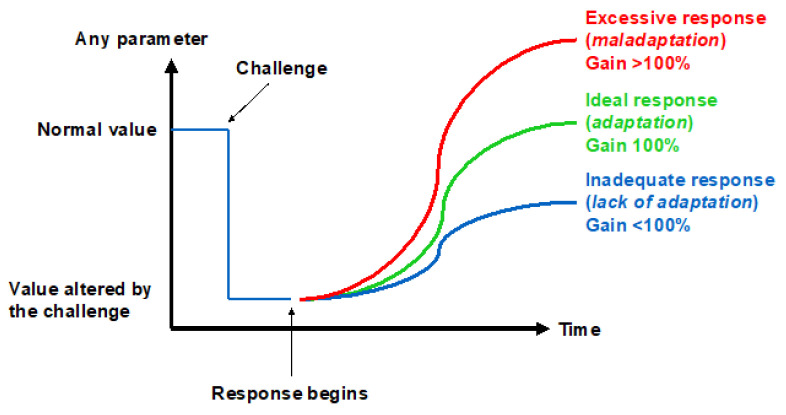
Three ways to respond to a challenge.

**Figure 5 ijms-24-03670-f005:**
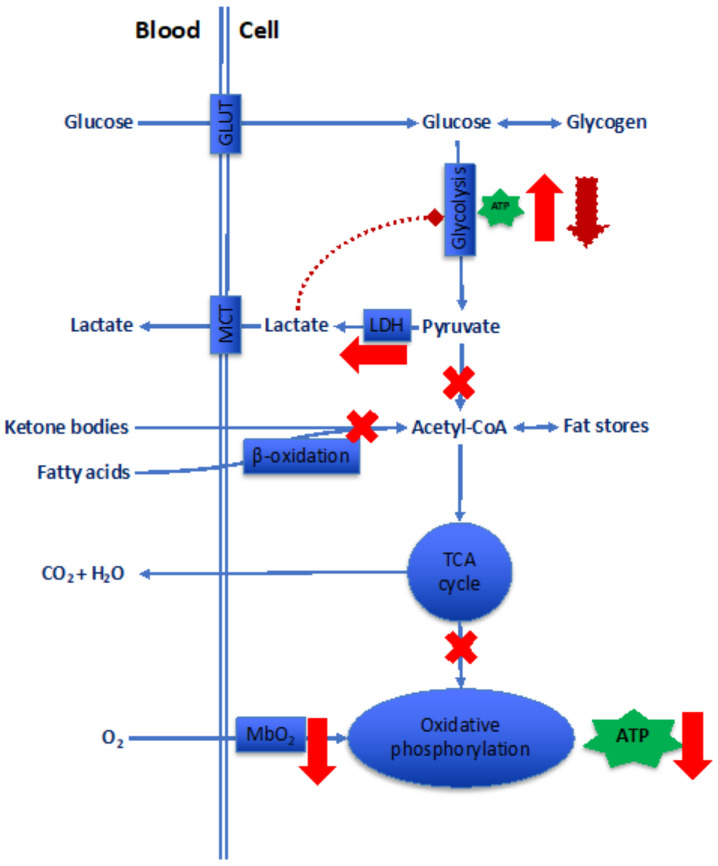
Scheme of the ATP production paths. In normoxia, most of the ATP is produced aerobically with a stringent need for O_2_ during oxidative phosphorylation, which follows the tricarboxylic acid (TCA) cycle. The utilization of O_2_ is helped by the presence of myoglobin (Mb) in contractile tissues. The oxidizable substrates that most contribute to aerobic ATP production are glucose, fatty acids, and (in heart and brain) ketone bodies. In acute hypoxia (red arrows and crosses), the O_2_ supply to oxidative phosphorylation is reduced along with the aerobic ATP production. Reduced disposal of acetyl-CoA by the TCA cycle, which is inhibited by hypoxia because of the hypoxia-induced downregulation of oxidative phosphorylation, in turn blocks the utilization of fatty acids (and ketone bodies in heart and brain). Simultaneous stimulation of glycolysis transiently increases anaerobic ATP production from glucose at the cost of increased lactate synthesis as partial compensation for the depressed aerobic ATP production. For longer hypoxia (brown dotted line), lactate overproduction is not compensated by proportional upregulation of the monocarboxylate transporters (MCT) at least in contractile tissues (see 3.1.4 Lactate shuttle). This leads to intracellular lactate buildup with lactate-driven inhibition of the glycolytic flux and impairment of anaerobic ATP production.

**Figure 6 ijms-24-03670-f006:**
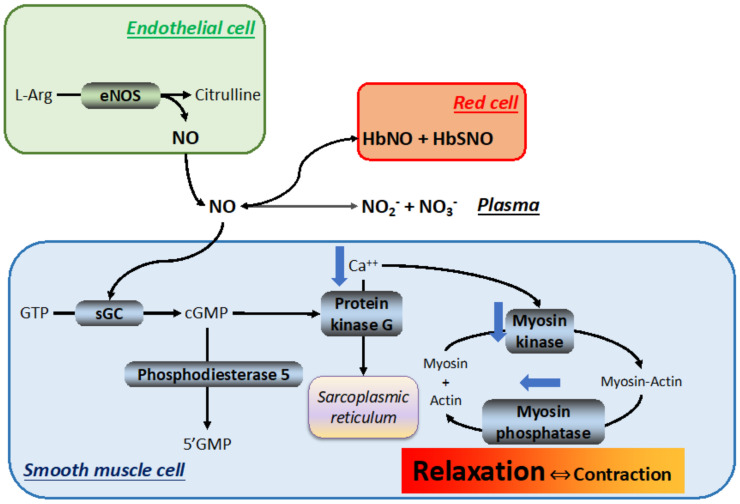
Simplified scheme of the mechanisms that modulate the NO–cGMP pathway in the smooth muscle cell. Four compartments are examined: the endothelial cells (green) that generate NO, the RBCs (red) that store NO as nitrosylated Hb (HbNO) or nitrosothiolated Hb (HbSNO), the plasma that stores NO as inorganic nitrites and nitrates (NO_2_^−^ + NO_3_^−^), and the smooth muscle cells (light blue). Here, NO favors the decrease in [Ca^++^] that is sequestered in the sarcoplasmic reticulum, diminishing the activity of myosin kinase and favoring muscle relaxation.

**Figure 7 ijms-24-03670-f007:**
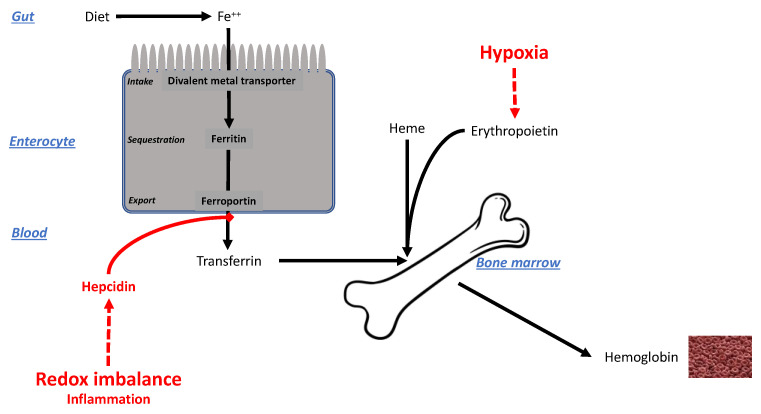
Scheme of iron handling and the impact of hypoxia and inflammation.

**Figure 8 ijms-24-03670-f008:**
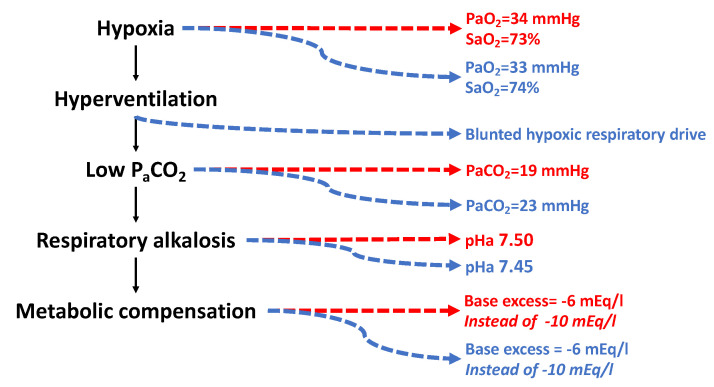
Respiratory adjustment to altitude (6450 m/9.3 %O_2_) in unacclimatized Caucasians (red) and acclimatized Sherpas (blue). Data from [212,264].

**Figure 9 ijms-24-03670-f009:**
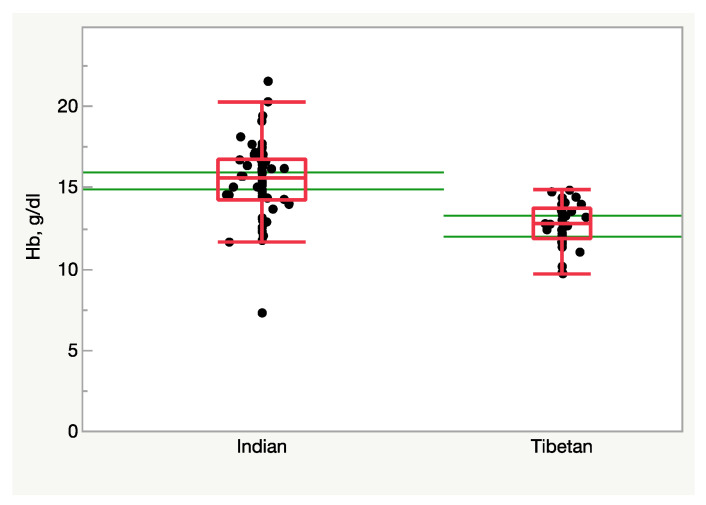
Hemoglobin concentration in the blood of Tibetan and Indian males (n = 37 and 56, respectively) residing at <500 m/>20.1 %O_2_. Tibetans display a lower Hb level (*p* < 0.0001, Student’s *t*-test). The green lines represent the 95% confidence limits.

**Table 1 ijms-24-03670-t001:** A few historical excerpts on the impact of hypoxia on mankind.

30 BC: First description of the effects of hypoxia by Tseen Han Shoo, while traveling along the Silk Road into Hindu Kush	”...the Great Headache Mountain, the Little Headache Mountain [… where] men’s bodies become feverish, they lose color, and are attacked with headache and vomiting...” [13]
1590: Padre Jose de Acosta, Spanish Jesuit, while crossing the Andes	“Many soldiers felt ill during the passage through Paso de Pariacaca” (4800 m/11.6 %O_2_), “… I was surprised by the so strange pain and deathful that I fell on the ground and noticed that air was so thin and not suitable for human breathing…”
1862: Henry Coxwell and James Glaisher, when flying at 8850 m at Wolverhampton	“.. Arms and legs became paralyzed, we felt blindness, difficulty in talking and experienced loss of conscience …” When back, they could recover fully.
1875: Sivel, Tissandier and Croce-Spinelli, in a balloon at 8500 m	They experienced loss of consciousness; Tissandier only recovered but became deaf.
1897: Angelo Mosso (1846–1910)	Published “Fisiologia dell’Uomo sulle Alpi” (Human Physiology in the Alps, Fratelli Treves Editors), where he described experiments performed at Capanna Regina Margherita, 4554 m/12.0 %O_2_, accounting for muscle strength, respiration, blood circulation, myocardial performance, nutrition, vital capacity, mountain sickness, and neurophysiological issues.
1916: Alexander Kellas (1868–1921)	In an article for the Royal Geographical Society, he analyzed the possibility of reaching the highest peaks of the Himalayas. In a later manuscript, which he failed to publish in life, he concluded that a physically gifted and well-trained man is able to climb Everest without O_2_ [14].
1953: Edmund Hillary and Tensing Norkay	First climb of Mt. Everest (8848 m/6.6 %O_2_) with the aid of supplemental O_2_.
1978: Reinhold Messner and Peter Habeler	First climb of Mt. Everest without the aid of supplemental O_2_.
1979–2022: Mt. Everest (8848 m/6.6 %O_2_)	Up to 270 climbers died, most from causes linked directly or indirectly to hypoxia.
1987: Robert M. Winslow (1941–2009) and Carlos Monge Cassinelli (1921–2006)	Published “Hypoxia, Polycythemia, and Chronic Mountain Sickness” (Johns Hopkins University Press, ISBN 0-8018-3448-1) in which they describe the origins of chronic mountain sickness in high-altitude natives in various parts of the world.

**Table 2 ijms-24-03670-t002:** Oxygen transport characteristics in Caucasians and Sherpas at varying altitudes. All data except P50 are from [212]. The P50 was calculated using the equations reported in [209] assuming carbon-monoxide-bound Hb = 0 and temperature = 37 °C.

	Altitude/%O_2_	Arterial pH	Arterial PCO_2_	Arterial PO_2_	[Hb]	DPG/Hb	P50
			mmHg	mmHg	g/L		mmHg
Caucasians	0/21.0	7.38	40	95	160	0.80	27.9
	3400/13.9	7.46	22	51	170	1.03	27.4
	5050/11.3	7.48	21	44	191	1.04	26.9
	6450/9.3	7.50	19	34	201	1.36	28.8
Sherpas	3400/13.9	7.46	29	53	169	1.01	27.5
	5050/11.3	7.45	27	42	186	1.04	27.9
	6450/9.3	7.45	22	33	NA	1.20	29.0

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
