# Peer review of "The Oxygen Cascade from Atmosphere to Mitochondria as a Tool to Understand the (Mal)adaptation to Hypoxia"

_ijms, 2023, doi:10.3390/ijms24043670_

Round 1

Reviewer 1 Report

Very interesting and informative review on hypoxia in general and to understand adaptation and maladaptation in particular. 

It appears to me that the authors would have also included high altitude work carried out  by many researchers on Yak, the most sturdy animal for high mountain regions and said to be genetically adapted to hypobaric hypoxia and cold. While going through these studies, some underlying physiological mechanism along with genomics and proteomics  would have provided a better scope for this review article.

Authors have very efficiently reviewed almost all the relevant domains of hypoxia research and thus deserve appreciation. However, the work on hypoxia research by  Nobel Laureates in Physiology and Medicine, 2019, William Kaelin., Peter Ratcliffe and Gregg Semenza would have given a better insight on cellular adaptation to hypoxia. 

Whatever volume of oxygen is inhaled by an individual, oxygen delivery ( DO2- the amount delivered at mitochondrial level ) holds an important role in energy generation. Authors may like to include this aspect also in their future research. How to calculate DO2 by a simple method other than the mathematical calculation holds significance for hypoxia research.

Author Response

Reply to all the Reviewers

We sincerely thank all the Reviewers for their positive judgment as well as the very valuable and thoughtful suggestions that have undoubtedly improved the quality of this manuscript. Here is the point-by-point reply to all their comments.

Reviewer 1

Very interesting and informative review on hypoxia in general and to understand adaptation and maladaptation in particular.

It appears to me that the authors would have also included high altitude work carried out  by many researchers on Yak, the most sturdy animal for high mountain regions and said to be genetically adapted to hypobaric hypoxia and cold. While going through these studies, some underlying physiological mechanism along with genomics and proteomics  would have provided a better scope for this review article.

Reply. We fully understand this concern of the Reviewer. As a matter of fact, we have long debated whether to insert animals’ data in this review and we finally opted to exclude the studies performed on animals as outlined at the end of the second paragraph in the Introduction and in the Limitations of the study. The main reasons for this choice include the fact that not only Yaks should be considered, but also Andean animals such as llamas and alpacas, as well as animals dwelling in the Tibetan plateau as certain types of moles, rats and more. This might have complicated the article to a considerable degree, as talking of genetic traits would be inconclusive if not accompanied by a discussion on the phenotypic effects of genetic markers at all the levels of the Oxygen Cascade. Perhaps the review would have doubled in length and, more critically, complexity. Therefore, although we admit that this represents a drawback in this manuscript, we kindly hope that the Reviewer will understand our opinion regarding this request. This point of view is now reported in the limitations of the study as it follows:

“Several geographically isolated high-altitude mammals such as yaks [304, 305], llamas [306], pikas [307] and many others have been considered models for the evolutionary adaptation to hypoxia. Comparison studies using such models may indeed constitute an appreciable experimental bench to test the credibility and suitability of the adaptation patterns observed in humans. However, each animal displays species-specific characteristics that would deserve separate analyses for each animal type and each genetic trait to enable meaningful comparison with the layouts observed in humans. Therefore, despite being aware of this limitation, in this review we opted to focus on data obtained in human populations, as well as in controlled in vitro and in vivo laboratory contexts, thereby excluding valuable studies conducted in altitude dwelling animals, especially for what concerns -omics studies.”

Authors have very efficiently reviewed almost all the relevant domains of hypoxia research and thus deserve appreciation. However, the work on hypoxia research by  Nobel Laureates in Physiology and Medicine, 2019, William Kaelin., Peter Ratcliffe and Gregg Semenza would have given a better insight on cellular adaptation to hypoxia.

Reply. We fully agree with the Reviewer. Indeed, the first paragraph under 3.1.1 which deals with HIF, already mentions the contribution given by the three Nobel Laureates. In the revised version, such contribution has been further developed:

Understanding how cells sense and adapt to O2 availability worth the 2019 Nobel Prize in Physiology and Medicine to William Kaelin, Peter Ratcliffe and Gregg Semenza, as well as robust worldwide reconnaissance. This ubiquitarian mechanism constitutes the O2 sensing function that orchestrates most, although not all, the cell responses to O2 shortage. The transcription factor family of hypoxia-inducible factors (HIF), a fundamental part of the O2 sensing system, exploits its function by inducing or repressing the expression of hundreds of genes encoding for proteins that target a relevant number of metabolic (e.g., glycolysis), morphological (e.g., angiogenesis), cell proliferation and survival (e.g., apoptosis, cell cycle control) and molecular (e.g., production of erythropoietin) processes.”

Whatever volume of oxygen is inhaled by an individual, oxygen delivery ( DO2- the amount delivered at mitochondrial level ) holds an important role in energy generation. Authors may like to include this aspect also in their future research. How to calculate DO2 by a simple method other than the mathematical calculation holds significance for hypoxia research.

Reply. We fully agree with the Reviewer, indeed the direct measurement of DO2 at the mitochondrial level may help great advancement in the studies related to hypoxia. Unfortunately, a method that enables this avoiding indirect mathematical calculations is still unavailable and may form a good target for future research. This concept is now introduced in the concluding remark of 3.1.3 (Bioenergetics):

“In summary, the bioenergetic changes associated with hypoxia, the logical consequences of the aerobic-to-anaerobic switch, may favor hypoxia adaptation, but the unavailability of methods to measure directly the O2 delivery in vivo at the mitochondrial level without the use of indirect mathematical calculations prevents appreciating how the bioenergetic changes may impact hypoxia adaptation. In addition, the role of pleiotropic factors that may modulate aerobic and anaerobic energy production as EPO, although being already hypothesized [89], is still to be worked out.”

Reviewer 2 Report

The review of Samaja & Ottolenghi describes the adaptation of the human body to hypoxia. It presents numerous data and is well organized. It is surely of interest for researchers of various fields but, in some instances, it looks overbiased toward “maladaptation”. See for examples lines 650-651.

Figure 3 is not correct : the oxygen does not go from venous blood to cells but from capillaries to cells, please modify.

Lines 174-175, why do the authors claim that ? please explain what it was meant.

Line 196 : the hydroxylation takes place on two prolines, not one.

Line 197 is not correct.

Line 244 : I do not agree, HIFs do play a clear tole in hypoxia adaptation

Line 247: there are non enzymatic processes that generate ROS in normal conditions (e.g. oxygen leakage from mitochondrial complexes I and III).

Line 254: in most cases, there are more ROS produced from complex III than from complex I.

Line 293: the sentence is not clear.

Lines 296, 311 and figure 5: this is not correct, the oxygen is not used in the TCA cycle but during the oxidative phosphorylations.

Line 317: why “failed” ?

Line 345: by which process ?

Line 372: this is a feeling not a scientific fact.

Line 427: it is not because there is no change in expression, that the transporter is not involved. Only intervention experiments (for ex through gene silencing) could answer to this question.

Line 477: why is that so ?

Line 480: which cell type, to what type of ECM ? how does hypoxia impairs cell adhesion?

Line 684: why “excessive” ?

Line 778 : which cell type ?

Line 996: please provide the name of the 4 genes

The English language needs to be checked, there are numerous errors (e.g. “an increase” in instead of “an increase of”).

There are also scientific language errors (e.g. “an increase in lactate”, “an increase in such or such protein” should be “an increase in lactate concentration/level”, an increase in such or such protein mRNA expression/protein abundance/activity”,…). “Hypoxia is present even at sea level”: not clear what it means. Hypoxia is not a lack of oxygen, it is a decrease in its concentration. Anoxia is a lack of oxygen.

Author Response

Reply to all the Reviewers

We sincerely thank all the Reviewers for their positive judgment as well as the very valuable and thoughtful suggestions that have undoubtedly improved the quality of this manuscript. Here is the point-by-point reply to all their comments.

Reviewer 2

The review of Samaja & Ottolenghi describes the adaptation of the human body to hypoxia. It presents numerous data and is well organized. It is surely of interest for researchers of various fields but, in some instances, it looks overbiased toward “maladaptation”. See for examples lines 650-651.

Reply. We have softened that paragraph that now reads:

“In summary, the observation that adapted populations exhibit blunted erythropoietic response, in contrast with exaggerated response often associated with the adverse clinical features observed in non-adapted populations, suggests that erythropoiesis might not be an adaptative response to hypoxia.”

Figure 3 is not correct : the oxygen does not go from venous blood to cells but from capillaries to cells, please modify.

Reply. The Reviewer is absolutely right. We have replaced “venous blood” with “capillary” in the figure and in the text.

Lines 174-175, why do the authors claim that ? please explain what it was meant.

Reply. If the Reviewer refers to the sentence “The human species was said to have the ability to adapt to hypoxia, although to a minor extent than other mammals do [27],” this is one of the messages of that article. However, we revised that sentence that now reads:

“The concept of adaptation physiology greatly depends on semantic problems related to the meaning of this term [27]. Although adaptation is believed to involve the establishment of compensatory responses to environmental challenges, not all the responses to a challenge may result to be adaptive.

Line 196 : the hydroxylation takes place on two prolines, not one.

Reply. Yes of course, the sentence was revised, and a reference was added:

“… the bulk of the control [is] entrusted to prolyl hydroxylases (PHD) and the factor inhibiting HIF (FIH) that, in the presence of O2, hydroxylate HIF at either the two prolyl or the asparaginyl residues, respectively (29).”

Line 197 is not correct.

Reply. The sentence “Hydroxylated HIF is then targeted to ubiquitination thereby subtracting HIF from gene expression control” has been amended in:

“When hydroxylated, HIF undergoes ubiquitination and can’t be active in its task of controlling the gene expression.”

Line 244 : I do not agree, HIFs do play a clear tole in hypoxia adaptation.

Reply. We recognize that the sentence “In summary, at present, it remains difficult to assess if HIFs may play significant roles in hypoxia adaptation” was inconsistent. In this version, that sentence has been amended:

“In summary, as HIFs may regulate the expression of several hundred genes, perhaps up to 2.6% of the human genome (48), it remains difficult at present to assess if all such  activities translate in useful hypoxia adaptation patterns, leaving the question whether HIFs participate to the adaptation processes relatively unanswered.”

Line 247: there are non enzymatic processes that generate ROS in normal conditions (e.g. oxygen leakage from mitochondrial complexes I and III).

Reply. We have corrected the statement:

“Besides fueling oxidative phosphorylation, O2 may also form highly reactive species (ROS) either through non-enzymatic (as mitochondrial uncoupling) or enzymatic (49) processes.

Line 254: in most cases, there are more ROS produced from complex III than from complex I.

Reply. We checked the literature and found that the matter whether the major ROS contributor under hypoxia is complex I or III is rather controversial. There are many variables that include tissue heterogeneity, duration and intensity of hypoxia and others. Thus, we opted to remove the part of the sentence that says, “especially at the level of mitochondrial complex I (51)”, which is irrelevant to this manuscript.

Line 293: the sentence is not clear.

Reply. The sentence “In summary, the challenge represented by oxidative stress is relevant in hypoxia-related contexts, but the building of the antioxidant defense, which would help the hypoxia adaptation process, might appear not fully exploited in several instances, which labels this potentially adaptive response as insufficient to meet its target” has been rewritten:

“In summary, there are two aspects to address the ox-redox challenge: the increased hypoxia-induced oxidative stress, and the buildup of endogenous antioxidant defenses, which might be not fully exploited especially for longer hypoxia durations. In terms of hypoxia adaptation, it might appear that the potentially adaptive response represented by the antioxidant defenses is insufficient to meet its target.”

Lines 296, 311 and figure 5: this is not correct, the oxygen is not used in the TCA cycle but during the oxidative phosphorylations.

Reply. Yes, of course, this crass error has been corrected in the beginning of 3.1.3, in the legend of Figure 5, in Figure 5 and elsewhere.

Line 317: why “failed” ?

Reply. We agree that the sentence “For longer hypoxia (brown dotted line), this lactate ultimately inhibits the flux through the glycolysis and impairs anaerobic ATP production due to failed upregulation of the monocarboxylate transporters (MCT) needed to export lactate and the acidity associated to this anion” was unclear. We rewrote that sentence as follows:

“For longer hypoxia (brown dotted line), lactate overproduction is not compensated by proportional upregulation of the monocarboxylate transporters (MCT) at least in contractile tissues (see 3.1.4 Lactate shuttle). This leads to intracellular lactate buildup with lactate-driven inhibition of the glycolytic flux and impairment of anaerobic ATP production. “

Line 345: by which process ?

Reply. The underlying mechanisms are now explained:

“…high lactate irreversible damage the O2 utilization paths by at least two mechanisms: (a) impairment of phosphorylation coupling efficiency due to the dissipation of the H+ gradient across the inner mitochondrial membrane via upregulation of the lactate-H+ co-transport, and (b) lactate-induced increased amplitude of the Ca++ transients that leads to increased intracellular Ca++ load with higher costs required for pumping out Ca++ (82).

Line 372: this is a feeling not a scientific fact.

Reply. That sentence has been rewritten as it follows:

“In summary, the bioenergetic changes associated with hypoxia, the logical consequences of the aerobic-to-anaerobic switch, may favor hypoxia adaptation, but the un-availability of methods to measure directly the in vivo O2 delivery at the mitochondrial level without the use of indirect mathematical calculations prevents appreciating how the bioenergetic changes may impact hypoxia adaptation. In addition, the role of pleiotropic factors that may modulate aerobic and anaerobic energy production as EPO, although being already hypothesized (89), is still to be worked out.”

Line 427: it is not because there is no change in expression, that the transporter is not involved. Only intervention experiments (for ex through gene silencing) could answer to this question.

Reply. The sentence “In summary, available data do not support a relevant role for MCT in hypoxia adaptation” has been rewritten:

“In summary, in the lack of targeted studies addressing the role of MCT to favor lactate export in somatic cells, there is little support for a relevant role for MCT in hypoxia adaptation.”

Line 477: why is that so ?

Reply. The paragraph has been completely rewritten to accommodate this Reviewer’s concern:

“Besides its main function of vasorelaxant to enhance tissue perfusion and oxygenation, NO restores peripheral blood mononuclear cell adhesion that is considerably reduced in hypoxia. This function is explicated via interaction with the mechanisms that affect remodeling and the extracellular matrix. First, hypoxia-induced overexpression of eNOS tends to rescue basal NO levels, which recovers cell-matrix adhesion via cGMP-dependent protein kinase [124]. This favors the cell ability to adhere to the extracellular matrix, an essential step to maintain normal homeostasis, which is hampered by hypoxia that reduces NO bioavailability (see below). In addition, while hypoxia-induced over-perfusion of the pulmonary bed destabilizes the endothelial cells by increasing their leakiness via depolymerization of F-actin stress fibers (which impairs the permeability of the pulmonary vascular endothelial bed potentially leading to high altitude pulmonary edema), activating the NO/cGMP pathway through NO donors or sildenafil can revert this process [125].”

Line 480: which cell type, to what type of ECM ? how does hypoxia impairs cell adhesion?

Reply. The entire paragraph has been rewritten to  meet the Reviewer’s concern, see above.

Line 684: why “excessive” ?

Reply. The Reviewer is right, we deleted that term.

Line 778 : which cell type ?

Reply. Epithelial cells, as outlined in the sentence:

“The thickness of the basal lamina, a layer of extracellular matrix formed by epithelial cells,...”

Line 996: please provide the name of the 4 genes

Reply. The four genes common in the three populations are: PIK3CB, HLA-DQB1, CNTNAP2, and DLG2. This is now added to the text:

“…with only 4 genes (PIK3CB, HLA-DQB1, CNTNAP2, and DLG2) commonly altered in all the populations”.

The English language needs to be checked, there are numerous errors (e.g. “an increase” in instead of “an increase of”). There are also scientific language errors (e.g. “an increase in lactate”, “an increase in such or such protein” should be “an increase in lactate concentration/level”, an increase in such or such protein mRNA expression/protein abundance/activity”,…).

Reply. We corrected all the misuses outlined by the Reviewer.

“Hypoxia is present even at sea level”: not clear what it means.

Reply. Right, modified in “Hypoxia may occur even…”

Hypoxia is not a lack of oxygen, it is a decrease in its concentration. Anoxia is a lack of oxygen.

Reply. Correct, “the lack of oxygen” was corrected everywhere in the manuscript.

Reviewer 3 Report

An interesting review article requires minor modifications. 

1. In the Fig. 6, check the type setting errors.

2. Similar to oxidative stress, nitrosative stress plays an important role in the pathology of various diseases during hypoxia. Discuss the relation between hypoxia and nitrosative stress.

3. Briefly describe how the cardiac, nervous and pulmonary tissues adapt initial hypoxic conditions in the context of pathophysiology.

Author Response

Reply to all the Reviewers

We sincerely thank all the Reviewers for their positive judgment as well as the very valuable and thoughtful suggestions that have undoubtedly improved the quality of this manuscript. Here is the point-by-point reply to all their comments.

Reviewer 3

An interesting review article requires minor modifications.

  1. In the Fig. 6, check the type setting errors.

Reply. We checked Figure 6 for typesetting errors.

  1. Similar to oxidative stress, nitrosative stress plays an important role in the pathology of various diseases during hypoxia. Discuss the relation between hypoxia and nitrosative stress.

Reply. We partially addressed this important topic in a dedicated paragraph on page 14. We agree that the topic of nitrosative stress is fundamental, thus we reformulated the following sentence: “Despite high NO levels are toxic as for the fact that NO is itself a free radical, in physiological or slightly supraphysiological concentrations NO protects the hypoxic body in several ways” as follows:

“High circulating NO levels are toxic because NO is itself a free radical and its reaction with O2 generates peroxynitrite, a most dangerous reactive nitrogen species. This results in nitrosative stress, which plays an important role in the pathology of various diseases such as heart failure [124]. On the other hand, when present at physiological or slightly supraphysiological levels, NO protects the hypoxic body in several ways.”

  1. Briefly describe how the cardiac, nervous and pulmonary tissues adapt initial hypoxic conditions in the context of pathophysiology.

Reply. An excellent review on this exciting topic has been very recently published in this Journal under the same Special Issue. Therefore, we rewrote the paragraph that follows Figure 3:

“The ability to adapt to environmental changes is a key feature in the evolution of the species. While the mechanisms underlying the cardiac, nervous and pulmonary tissues initial adaptation, or acclimatization, to hypoxic conditions have been recently worked out [1], the rules for recognizing the onset of long-term adaptation should include the appearance of complex characters that are too well fitted to the environment for the fit to have arisen by chance, and that help their bearers to survive and reproduce.”

  1. Mallet, R. T.; Burtscher, J.; Pialoux, V.; Pasha, Q.; Ahmad, Y.; Millet, G. P.; Burtscher, M., Molecular Mechanisms of High-Altitude Acclimatization. Int J Mol Sci 2023, 24, (2), 10.3390/ijms24021698.

Round 2

Reviewer 2 Report

The authors adequately addressed the comments.